# A central CRMP complex essential for invasion in *Toxoplasma gondii*

**Mirko Singer**[1,2]*, **Kathrin Simon**[1], **Ignasi Forné**[3], **Markus Meissner**[1]*

**1** Faculty of Veterinary Medicine, Experimental Parasitology, Ludwig-Maximilians-University (LMU) Munich, Germany, **2** Integrative Parasitology, Center for Infectious Diseases, Heidelberg University Medical School, Heidelberg, Germany, **3** Faculty of Medicine, Protein Analysis Unit, Biomedical Center (BMC), Ludwig-Maximilians-University (LMU) Munich, Martinsried, Germany

* Mirko.Singer@gmail.com (MS); Markus.Meissner@lmu.de (MM)

**Data Availability Statement:** All relevant data is within the paper and its supporting information files. The raw files of the mass spectrometry dataset is available at https://www.ebi.ac.uk/pride/

## Abstract

Apicomplexa are obligate intracellular parasites. While most species are restricted to specific hosts and cell types, *Toxoplasma gondii* can invade every nucleated cell derived from warm-blooded animals. This broad host range suggests that this parasite can recognize multiple host cell ligands or structures, leading to the activation of a central protein complex, which should be conserved in all apicomplexans. During invasion, the unique secretory organelles (micronemes and rhoptries) are sequentially released and several micronemal proteins have been suggested to be required for host cell recognition and invasion. However, to date, only few micronemal proteins have been demonstrated to be essential for invasion, suggesting functional redundancy that might allow such a broad host range. Cysteine Repeat Modular Proteins (CRMPs) are a family of apicomplexan-specific proteins. In *T. gondii*, two CRMPs are present in the genome, CRMPA (TGGT1_261080) and CRMPB (TGGT1_292020). Here, we demonstrate that both proteins form a complex that contains the additional proteins MIC15 and the thrombospondin type 1 domain-containing protein (TSP1). Disruption of this complex results in a block of rhoptry secretion and parasites being unable to invade the host cell. In conclusion, this complex is a central invasion complex conserved in all apicomplexans.

## Introduction

The apicomplexan parasite *Toxoplasma gondii* is capable to actively invade every nucleated vertebrate cell. How the parasite is capable to achieve such an extraordinary broad host range is unknown. While it has been suggested that the parasite expresses a large repertoire of receptors derived from unique secretory organelles (micronemes) that can bind to a great variety of surface markers [1], most micronemal proteins can be deleted without significantly affecting invasion in vitro. It is possible that the parasite can detect highly conserved surface proteins, glycosylated proteins, or even more conserved properties of cells like surface charge or membrane components of the plasma membrane. Each invasion event requires the discharge of the rhoptries, specialized secretory organelles found in all invasive stages of apicomplexans, and the establishment of the junctional complex, through which the parasite invades [2]. Since this

Project accession numbers: PXD031649 and PXD035654.

**Funding:** This study was supported by the DFG Equipment grant INST 86/1831-1. M.S. was funded by the Volkswagen Foundation by "Experiment!", A127931. The funders had no role in study design, data collection and analysis, decision to publish, or preparation of the manuscript.

**Competing interests:** The authors have declared that no competing interests exist.

**Abbreviations:** CLAMP, claudin-like apicomplexan microneme protein; CRMP, Cysteine Repeat Modular Protein; IFA, immunofluorescence assay; hyperLOPIT, hyperplexed localization of organelle proteins by isotope tagging; PAE, predicted alignment error; TM, transmembrane domain.

step is conserved in all apicomplexan parasites, we hypothesized the presence of a central protein complex, acting upstream of rhoptry secretion and being required for invasion into different host cells.

Host cell invasion is a stepwise process [3]. Upon initial contact with the host cell, micronemes are secreted. The micronemes contain a huge variety of micronemal proteins that also appear to function in a stepwise process from gliding motility, host cell recognition, activation of rhoptry secretion, and force transmission during parasite entry. Surprisingly, only few micronemal proteins have been demonstrated to be critical for host cell invasion, which might be due to redundancies within the arsenal of micronemal proteins, as suggested for AMA1 [4] [5]. Signaling cascades resulting in secretion of micronemes are conserved between *Toxoplasma* and *Plasmodium* [6]. The same is true for the main components of the gliding and invasion machinery [7].

In the case of *T. gondii*, the micronemal protein MIC8 appears to act upstream of rhoptry secretion [8] and it was suggested to be a receptor triggering secretion of the rhoptries. However, to date, neither interaction partners, nor host cell ligands could be identified that specifically interact with this protein, making its exact function enigmatic. Intriguingly, although crucial for host cell invasion by *Toxoplasma*, no direct homolog can be found in *Plasmodium*. The related protein MIC7 was recently characterized, and is important, but not essential for invasion of *T. gondii* [9]. Again, no direct homolog of MIC7 can be identified in *Plasmodium*.

Recently, a genome-wide screen allowed the identification of the claudin-like apicomplexan microneme protein (CLAMP) that also appears to have a role in rhoptry secretion and formation of the tight junction through which the parasite invades [10]. Unlike MIC7 and MIC8, CLAMP is highly conserved in apicomplexans and therefore might play a central role in the initiation of rhoptry secretion. Furthermore, comparative analysis of specialized secretory organelles across the superphylum of alveolates led to the identification of unique structures that appear to be conserved, such as a central fusion rosette that was identified in *Paramecium* [11], *Tetrahymena* [12], and later also *Coccidia*, leading to the speculation that these organelles are sharing the same evolutionary history [13]. Analysis of secretory mutants of *Paramecium tetraurelia* led to the identification of Nd proteins, such as Nd6 or Nd9 that are required for rosette formation and trichocyst secretion [14,15]. Interestingly, the armadillo repeat protein Nd9p [16] and RCC1-like protein Nd6p [17], identified in *Paramecium*, turned out to be highly conserved in apicomplexan parasites, where they appear to play the same, conserved role in rosette formation and rhoptry secretion [18].

More recently, the rhoptry secretion system has been characterized with electron tomography in multiple species of apicomplexa [19–21], but integration of this structural information and single protein function is still in its infancy.

In *Plasmodium*, stage-specific invasion factors and their receptors have been identified. In merozoites of *Plasmodium falciparum*, the essential complex of RH5, RIPR, and CyRPA is binding basigin on the erythrocytes surface via RH5 to trigger rhoptry secretion [22]. In *Plasmodium berghei* sporozoites, both CD36 and CD52 are essential for invasion into hepatocytes, binding CD81 [23]. Of all these, RH5 is restricted to the Laveranian subgenus in *Plasmodium* [24]; all other members have no homologs outside of *Plasmodium*. The switch from semiselective to cell type-selective invasion in the evolution of haemosporidia [25] suggests that restriction to a limited host cell repertoire is a secondary trait. Again, this would predict that species and stage-specific receptors are modulating a central invasion complex.

In *P. berghei*, four Cysteine Repeat Modular Proteins (CRMPs) have been shown to have diverse phenotypes in the mosquito stage, including oocyst egress, salivary gland invasion, and hepatocyte invasion [26,27]. Individual gene deletion results in mild growth phenotypes of blood stage parasites in both *P. berghei* and *P. falciparum* [28,29].

Here, we focused on the function of CRMPs in *T. gondii*, since both copies were predicted to be essential in a genome-wide screen [10], are predicted to be multiple transmembrane domain (TM) proteins, and form a separate cluster in a recently performed hyperplexed localization of organelle proteins by isotope tagging (hyperLOPIT) analysis [30].

We performed a bioinformatic analysis and identified CRMPs as a family of apicomplexan-specific proteins that are widely conserved within the phylum. Both CRMPs are specifically required for rhoptry secretion and subsequent host cell invasion. Finally, we show that CRMPs are present in a multi-subunit complex with at least two additional, essential micronemal proteins, MIC15 and TSP1, which are required for rhoptry secretion and invasion.

## Results

### The family of CRMPs in apicomplexan parasites

We performed a phylogenetic analysis of apicomplexan CRMPs and looked at the domain architecture (Figs 1A, 1B and S1). As local conservation is very low and many CRMPs have long insertions (S1C Fig), we defined a "conserved core," which includes two EGF-like domains and the transmembrane domains. Alignment of this core region resulted in comparable results (see S1A Fig). The phylogenetic tree suggests that CRMPA of Coccidia groups with CRMP1 and CRMP2 (the CRMPA clade) of Haemosporidia and CRMPB groups with CRMP3 and CRMP4 (the CRMPB clade) of Haemosporidia (Fig 1A). We could find two putative CRMPs in Cryptosporidia but only one CRMP in the recently sequenced marine Gregarine species *Porospora gigantea* [31], just as in *Gregarina niphandrodes*. Interestingly, in both *Theileria annulata* and *Babesia bovis*, CRMP1 and CRMP2 are next to each other on the same chromosome, with a central DNA sequence that is identical in both, suggesting a conserved gene duplication event predating their split into separate species. Recent data from *B. bovis* suggest that one gene of each group (CRMPA clade and CRMPB clade) is expressed in bloodstages and the other one in kinetes [32]. In *P. berghei*, single gene deletions of all four CRMPs have been generated [26,27], and in *P. falciparum*, phenotypic scores (Mutant fitness score) of (−0.19) to (−1.59) have been assigned [29]. In *Chromera velia*, a free-living apicomplexan ancestor, many diverse CRMPs can be found [33,34]. Together, this analysis demonstrates a huge diversity of apicomplexan CRMPs, with the occurrence of independent gene losses and duplications. However, at least one copy of CRMPA clade and CRMPB clade appears to be maintained throughout the evolution of most apicomplexans parasites. The two CRMPs of *T. gondii*, TGGT1_261080 and TGGT1_292020, were identified as critical for the lytic cycle of the parasite in a genome-wide CRISPR/Cas9 screen with a phenotypic score of −3.95 and −3.64, respectively [10]. Both proteins contain several cysteine-rich domains and 9 transmembrane domains. CRMPA also contains a Kringle domain that has been shown to be involved in protein–protein interaction [35] (Fig 1B).

### CRMPA and CRMPB show a unique subcellular localization

For localization of CRMPs, we used Cas9-YFP assisted selection marker free tagging in the recipient strain ΔKu80-DiCre [37] (S2 Fig). We generated two parasites lines with both proteins C-terminally tagged (RH-CRMPA-3HA[floxed] CRMPB-sYFP2 and RH-CRMPB-3HA[floxed] CRMPA-sYFP2) (Figs 2A and S4). The proteins appear to localize throughout the secretory pathway of the parasite with considerable retention time in the ER, with a clear accumulation close to the apical conoidal micronemes but distinct from the micronemes associated with the subpellicular microtubules where no significant colocalization with MIC2 or MIC8 [38] was evident (Fig 2B). In good agreement, hyperLOPIT (Localization of Organelle Proteins by Isotope Tagging) [30] predicts them as micronemal but places them in clusters next to the ER, distinct from other micronemal proteins (TAGM-MAP is micronemal with probability 0,

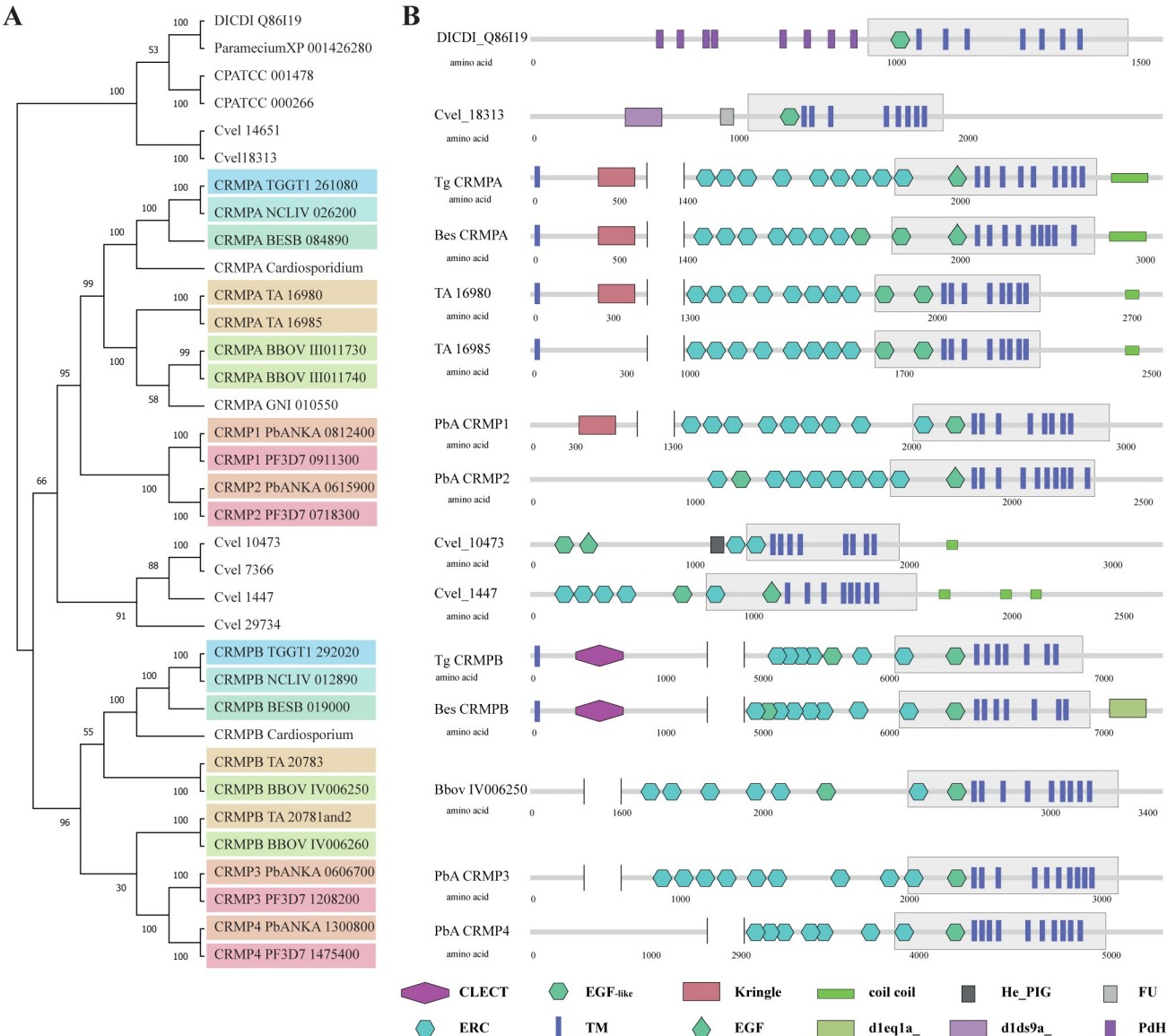

**Fig 1. Phylogeny and domain architecture of Cysteine Repeat Modular Proteins.** (**A**) Phylogenetic relationship of CRMPs is shown after alignment of the entire coding sequence, bootstrap analysis (100×) supported most branches, colored by apicomplexan species. For sequences and alignment, see S1 Table. (**B**) Domain architecture of selected CRMPs. Domains were predicted with SMART [36]. The conserved core containing the transmembrane domains was defined after the amino acid alignment and is indicated with a gray box. Some large N-terminal parts containing only internal repeats are not shown. Several domains are indicated: CLECT (c-type Lectin or carbohydrate recognition domain), ERC (Ephrin-receptor like), EGF-like (Epidermal growth factor-like), TM (transmembrane domain), Kringle (Kringle domain), EGF (Epidermal growth factor), coil coil (alpha helical coil coil domains), d1eq1a_ (apolipophorin III), He_PIG (putative Immunoglobulin-like fold), d1ds9a_ (outer arm of dynein light chain), FU (Furin-like repeats), PdH1 (parallel beta helix repeat). Additional data, see S1 Data.

MAGM-MCMC is micronemal with probability 1) (S1B Fig). The localization throughout the secretory pathway of the parasite was confirmed by colocalization with additional micronemal, Golgi, and ER markers (see below). Time lapse analysis of parasites expressing Halo-labeled CRMPs (RH-CRMPB-Halo CRMPA-3HA^floxed and RH-Halo-CRMPA-3HA^floxed CRMPB-sYFP2) support a dynamic localization throughout the secretory pathway (S1 Movie).

Next, we wished to analyze the behavior of CRMPB and CRMPA within the secretory pathway and during host cell invasion and inserted an additional tag in the extracellular part of the

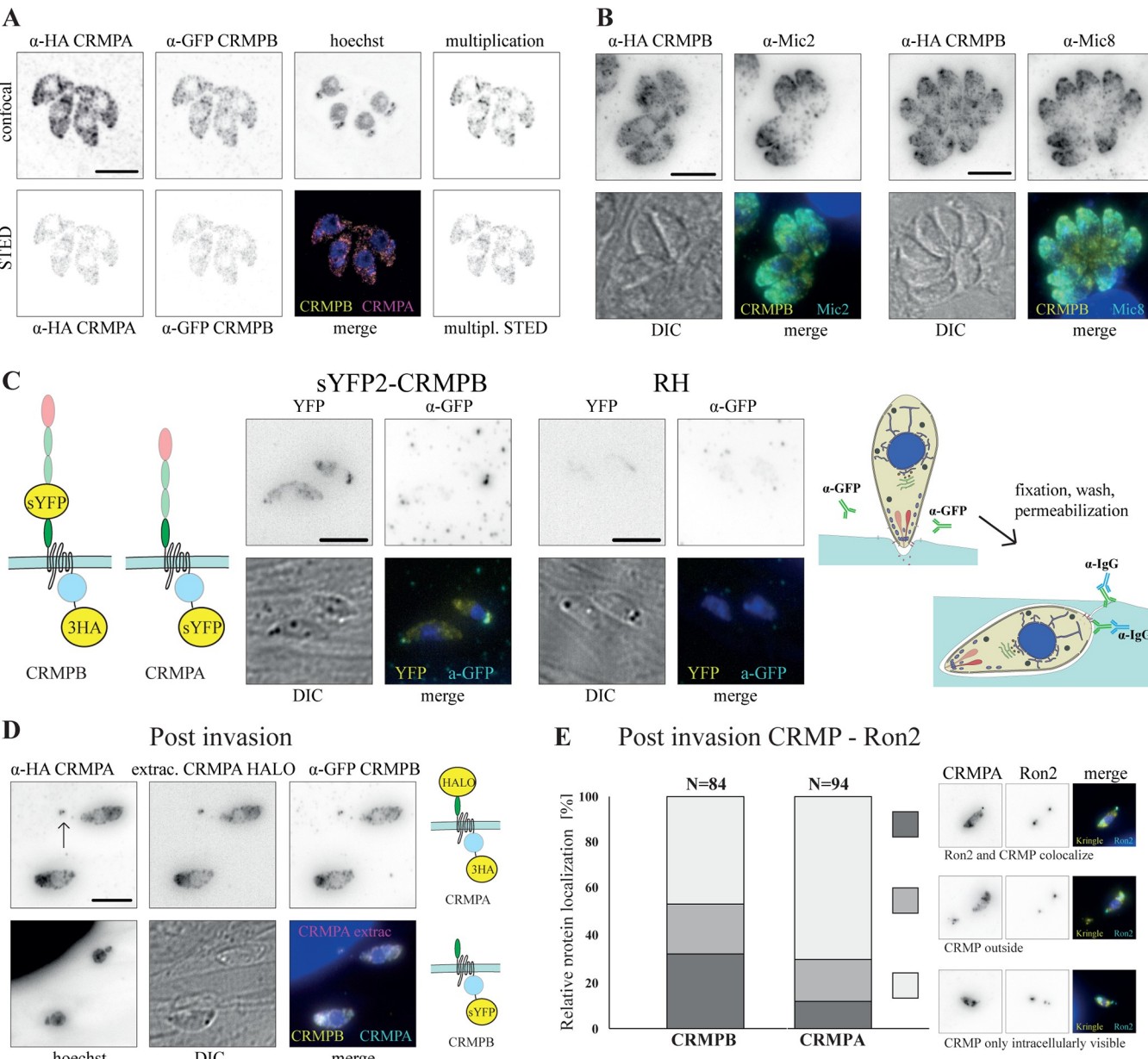

**Fig 2. Subcellular localization of CRMPB and CRMPA using STED microscopy.** (**A**) Colocalization of CRMPA and CRMPB. RH-CRMPA-3HA^floxed CRMPB-sYFP2 was stained with for HA (Star635P) and sYFP2 (Atto594). A colocalization map using the multiplication of both signals shows a high degree of colocalization within the whole cell. A parasite line with swapped tags shows comparable localization (S4 Fig). (**B**) Colocalization of RH-CRMPB-3HA^floxed stained for HA (Star635P) with micronemal proteins MIC2 and MIC8. (**C**) A pulse invasion assay of parasites with extracellular tagged CRMPB (RH-sYFP2-CRMPB-3HA^floxed CRMPA-sYFP2) was performed in the presence of α-GFP antibodies. Post-invasion parasites were fixed, permeabilized, and the a-GFP antibody was visualized. See schematics of experimental design on the right. YFP signal is all CRMPs of the parasite. (**D**) Parasites with extracellular CRMPA Halo (RH-Halo-CRMPA-3HA^floxed CRMPB-sYFP2) were labeled with Halo-Janelia 646 and fixed after pulse invasion in the presence of proponol. The putative invasion site is marked with an arrow. Tagging schematics of experimental design on the right. (**E**) Pulsed invasion of RH-CRMPB-3HA^floxed CRMPA-sYFP2 RON2-SNAP and RH-CRMPA-3HA^floxed CRMPB-sYFP2 RON2-SNAP after labeling with SNAP-Cell. Relative position of the basal RON2 signal at the site where the parasitophorous vacuole closes and the CRMP signal were scored as colocalization, CRMP outside of RON2, or only internal CRMP signal (see also S1 Data). Scale bars are 5 μm.

protein (S2 Fig). To determine if CRMPB is delivered to the plasma membrane, we analyzed parasites expressing CRMPB with an extracellular sYFP2 tag (RH-sYFP2-CRMPB-3HA^floxed CRMPA-sYFP2). For this, parasites were inoculated on host cells in the presence of a GFP

antibody that was visualized post-fixation after permeabilization of the parasite, resulting in staining of CRMPB that was at the plasma membrane during the experiment. (Fig 2C). This analysis revealed that in parasites that had recently invaded, CRMPB accumulates near the basal pole of the parasite or in the proximity, potentially the site of invasion. These data suggest that CRMPB is secreted to the plasma membrane.

Furthermore, analysis of freshly invaded parasites expressing double-tagged CRMPA and tagged CRMPB (RH-Halo-CRMPA-3HA$^{floxed}$ CRMPB-sYFP2) revealed an accumulation of CRMPB and CRMPA near the basal end of the parasite irrespective if the protein is tagged C-terminally (cytosolic tail) or if the tag has been placed in the extracellular part of the protein (Figs 2D, S4A, S4C and S4D), suggesting that these proteins are accumulating at the tight junction and are consequently excluded from the parasitophorous vacuole membrane during invasion. We also analyzed the localization of CRMPs of freshly invaded parasites relative to the tight junction using RON2, tagged with SNAP [2]. In some cases, we found CRMPs at or outside of the site marked by RON2 where the tight junction closed (Fig 2E). Live imaging (S2 Movie) suggests that CRMPB and CRMPA are present mainly at the basal end of moving and invading tachyzoites. A weaker signal was also observed at the apical tip in some instances. We have, however, no indication which of the observed localizations is functional for invasion. In one instance, the ring formation at the tight junction is visible, suggesting that CRMPs are excluded at the tight junction. We also observed trail deposition in RH-Halo-CRMPA but not in c-terminally labeled RH-CRMPB-Halo. This could be an indication that in a subset of the plasma membrane fraction, the extracellular part is proteolytically cleaved.

This relocalization of CRMPs is similar to what has been observed for other membrane-tethered proteins, such as MIC2, MIC8, or CLAMP [1,8,10].

## CRMPs are essential for invasion

For the functional characterization of CRMPB, we used the DiCre-System [39] that allows conditional excision of a gene of interest. In a first attempt, we inserted a LoxP site up- and downstream of CRMPB. However, gene excision rate was too low to reliably characterize the phenotype. Given the enormous size of CRMPB (27 kbp), we concluded that excision of the full-length gene via DiCre is inefficient. We previously inserted a LoxP site directly into the coding region of a gene (the MIC8 extracellular floxed parasite strain used as a positive control in this study (S2D Fig)). Inspired by work in *P. falciparum* [40], we decided to use an existing intron of CRMPB to insert a LoxP site. As branchpoints are generally rather close to the acceptor site at the 3′ end of the intron [41–43], we inserted the LoxP site 58 bp downstream of the 5′-end of the intron (S2A Fig). Induction with rapamycin resulted in efficient excision of three cysteine repeats and all transmembrane domains of CRMPB, as evidenced by plaque assay, immunofluorescence assay (IFA), and confirmation of correct excision by genotyping (Figs 3A, 3D and S2).

In the case of CRMPA, we also introduced the LoxP site into an intron (S2B and S2E Fig). Here, the last two transmembrane domains and the C-terminal tail are disrupted upon DiCre induction. Similarly to CRMPB, we achieved efficient excision upon treatment with rapamycin. Disruption of either gene resulted in parasites incapable of forming any plaques in an HFF monolayer, similar to positive control parasites (MIC8 extracellular domain). For both CRMPs, insertion of the LoxP site did not result in any observable fitness loss or mislocalization (Figs 2A, 3A and S4B). We could still detect residual protein after 48 h, and invasion was only slightly inhibited. In contrast, 72 h after induction, both proteins are undetectable in immunofluorescence assays (Fig 3D). Consequently, all assays presented here were performed 72 h after induction.

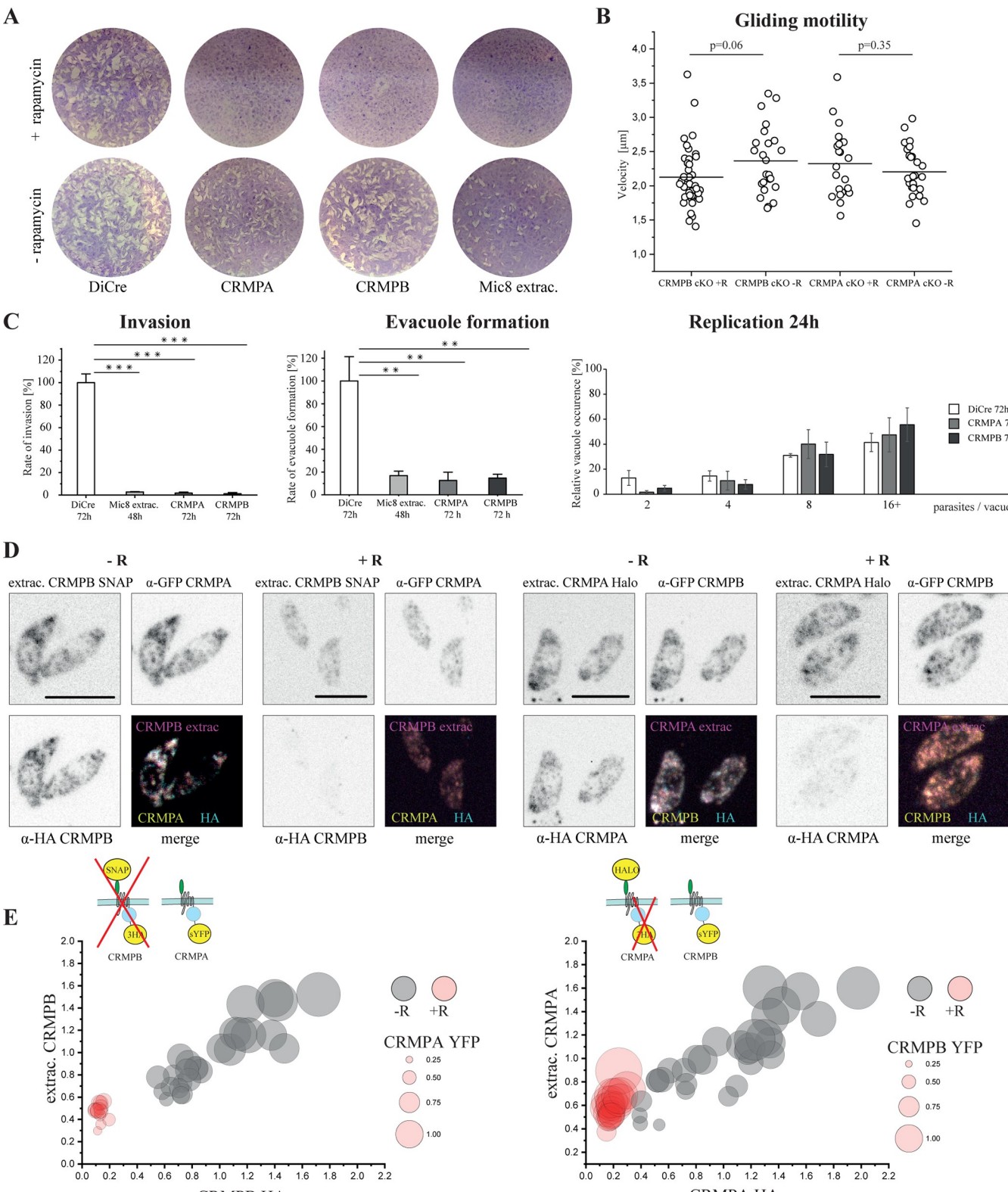

**Fig 3. Phenotypical analysis of CRMPs.** (**A**) Plaque formation after 7 days of CRMPB and CRMPA, both with LoxP in the Intron (S2 Fig). MIC8 floxed extracellular domain (RH-MIC8extrac.^floxed) is shown as positive control. Representative plaque assays are shown. (**B**) Gliding motility was assessed on FBS-coated dishes 3 days post-induction using rapamycin. The speed of tachyzoites moving for 10 or more consecutive seconds is shown. Statistical analysis: two-

tailed Student *t* test. (**C**) Invasion, evacuole formation and intracellular replication was measured 72 h post-induction (48 h for MIC8 control). Standard deviation between three independent replicates is shown. Statistical analysis: two-tailed Student *t* test. ** = $p < 0.01$. *** $p < 0.001$. (**D**) Localization of triple tagged CRMPs (RH-SNAP-CRMPB-3HA$^{floxed}$ CRMPA-sYFP2 and RH-Halo-CRMPA-3HA$^{floxed}$ CRMPB-sYFP2) ±rapamycin is shown after 72 h. Extracellular parasites are imaged to avoid host cell background in the analysis. Channels are normalized to −rapamycin. Note that the 3HA tag is excised after rapamycin induction, whereas the extracellular tag of CRMPA is not (see S2 Fig) and tagging schematics below, also indicating the part of the protein excised on the genomic level. (**E**) Tachyzoites shown in (**D**) are analyzed for total fluorescence. Each circle represents one tachyzoite; the color indicates ±rapamycin induction. Position on the x-axis indicates relative 3HA fluorescence; position on the y-axis the relative presence of the extracellular tag and circle size the relative presence of the other CRMP (sYFP2). Each measurement is normalized to the average intensity within all tachyzoites analyzed of the no rapamycin group. Statistical analysis: two-tailed Student *t* test. All comparison of + and −rapamycin are $p < 0.0001$, except CRMPB YFP ±R: $p = 0.23$. Scale bars are 5 μm. All raw data in S1 Data.

CRMPB and CRMPA did not have any measurable impact on gliding motility (Fig 3B). We also performed a micronemal secretion assay and could not observe any difference compared to DiCre (S3A Fig). Nevertheless, host cell invasion was severely reduced compared to wild type (Fig 3B and 3C). Since this phenotype appeared similar to the phenotypes observed for CLAMP and MIC8, where parasites are unable to secrete their rhoptry content and consequently fail to establish a tight junction, we analyzed evacuole formation. Indeed, evacuole formation was significantly reduced, suggesting a block in invasion prior to rhoptry secretion. In contrast, intracellular development was not affected (Fig 3C). In conclusion, CRMPA and CRMPB appear to act at a similar time during the invasion pathway as MIC8 and CLAMP. Analysis of the localization of RON2-SNAP post-invasion also supported a role of CRMPs in rhoptry secretion (S6 Fig). We only observed a single invasion event each of parasites depleted of CRMPA and CRMPB that showed normal RON2-SNAP localization (rhoptry staining and signal at the basal pole). All other parasites remained extracellular and RON2-SNAP localization was within the tachyzoites, suggesting that they did not secrete any rhoptry content. We therefore speculated that CRMPA and CRMPB might act together as part of a multi-subunit complex that is required to trigger rhoptry secretion.

Many protein complexes are destabilized once one component is removed. Therefore, we analyzed the interdependence of CRMPA and CRMPB by investigating their respective localization and relative expression levels upon disruption of the putative partner (Fig 3D and 3E). We measured total fluorescent intensity of individual parasites analyzing protein abundance with and without excision of the Ha-tagged protein (Fig 3E). Removal of CRMPB results in significant lower abundance of CRMPA. In contrast, removal of CRMPA had only slight effects on the protein levels of CRMPB. This suggests that the presence of CRMPB is required for CRMPA, but not vice versa. Potentially, this discrepancy can be explained by expression of residual protein after rapamycin induction. The positioning of the intron results in loss of all TMs for CRMPB, but only the last two TMs of CRMPA. Together, these data indicate that both proteins interact. We therefore decided to perform a detailed analysis of this putative complex.

## CRMPA and CRMPB are in a complex with at least two additional invasion factors

To test if both CRMPs are interacting and form a complex and to identify additional interaction partners, we performed mass spectrometry (Fig 4). First, we performed pulldown of RH-CRMPB-3HA$^{floxed}$ CRMPA-sYFP2, RH-CRMPA-3HA$^{floxed}$ CRMPB-sYFP2, and DiCre parasites using magnetic beads coated with α-Ha antibodies. Only few proteins were enriched in both pulldowns compared to the wild-type control, MIC15 (TGGT1_247195), and both CRMPs themselves (Fig 4A, 4B and 4D). To achieve a higher sensitivity, we tagged the CRMPs with TurboID at the C-terminus (RH-CRMPB-TurboID and RH-CRMPA-TurboID) [44]. Biotin labeling was performed for 30 min on extracellular tachyzoites (Fig 4C and 4E). Both

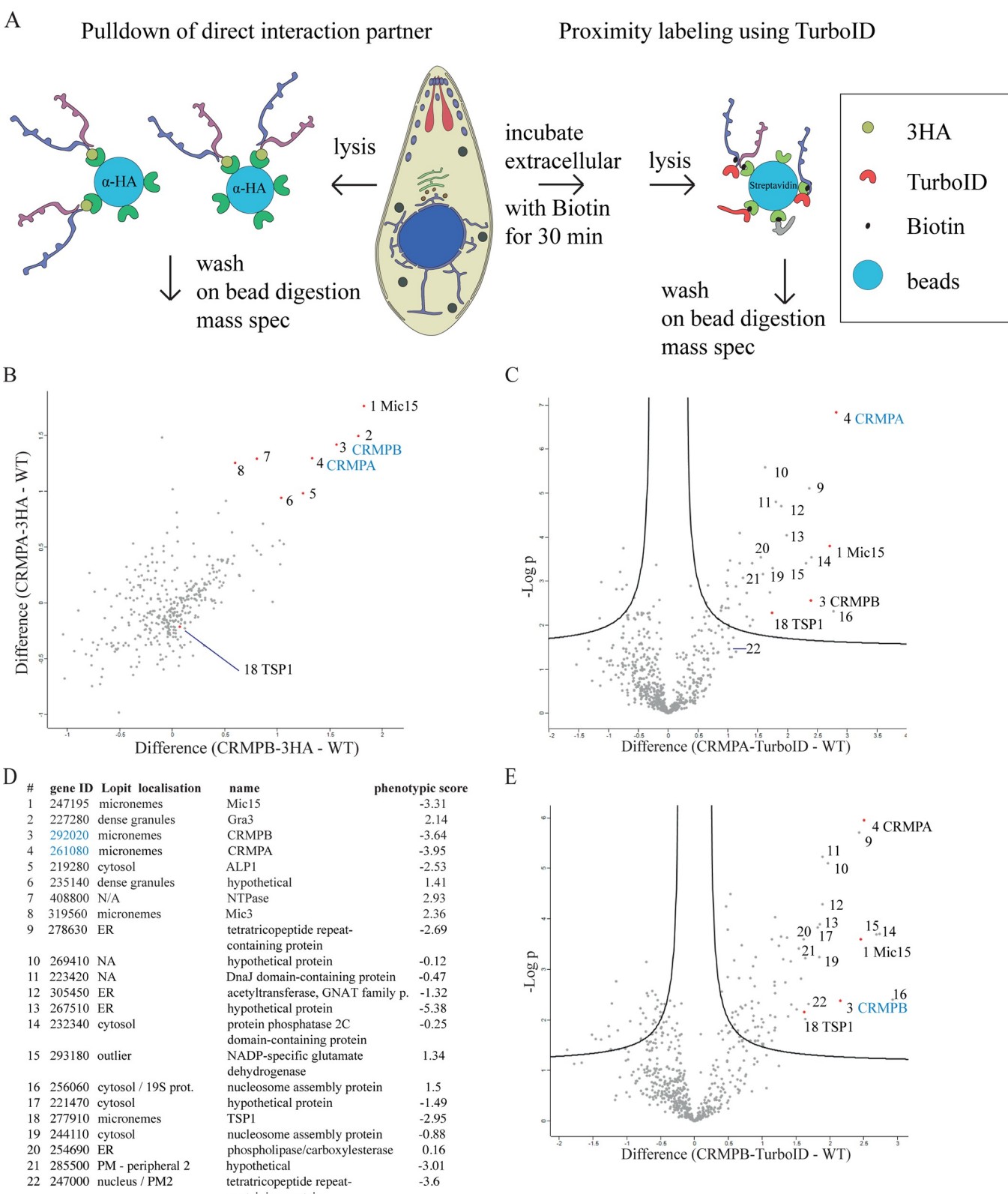

**Fig 4. Identification of interaction partners.** (**A**) Experimental outline of mass spectrometry–based protein identification. Parasites with C-terminally tagged 3HA were used for direct pulldown experiments (depicted on the left, results shown in (**B**)). In a separate experiment, C-terminally tagged CRMPA or B with TurboID was used in an attempt to identify more transient interaction partners of the CRMPs post-secretion. Here, pulldown with streptavidin-coupled beads

was performed after incubation of extracellular tachyzoites with biotin for 30 min. (**B**) Interactors of CRMPA 3HA and CRMPB 3HA. The normalized relative log 2 protein intensity of CRMPB 3HA—WT are blotted on the x-axis, while blotted on the y-axis for CRMPA-3HA—WT. Baits are highlighted in blue. Correspondence between numbers in the plot and protein names is listed in (**D**). (**C**) Transient interactors of CRMPA. Proteins enriched in the CRMPA TurboID sample are present in the upper right area of the graph. Protein tagged with TurboID is highlighted in blue. (**D**) List of proteins highlighted in (**B**), (**C**)**,** and (**E**). (**E**) Transient interactors of CRMPB. Proteins enriched in the CRMPB TurboID sample are present in the upper right area of the graph. Protein tagged with TurboID is highlighted in blue. For complete datasets, see S2 Data.

TurboID datasets show a very similar pool of proteins enriched in comparison to wild-type tachyzoites, supporting a stable complex formation by both CRMPs. In good agreement with the pulldown, CRMPB, CRMPA, and MIC15 were identified as top hits, with several additional proteins being enriched (Fig 4D). Based on their prediction by Lopit as plasma membrane-associated (plasma membrane peripheral 2 for TGGT1_247000 and TGGT1_285500) or micronemal (MIC15 and TSP1) and their negative phenotypic score, we shortlisted the tetratricopeptide repeat-containing protein (TGGT1_247000), thrombospondin type 1 domain-containing protein (TSP1, TGGT1_277910), a protein with three TSP1 repeats (MIC15, TGGT1_247195), and the uncharacterized protein with four transmembrane domains (TGGT1_285500) for further characterization. MIC15 and TSP1 are predicted to contain a single transmembrane domain; TSP1 lacks an N-terminal signal peptide. The protein TGGT1_247000 contains a single N-terminal transmembrane domain with the rest of the protein predicted to be noncytosolic; 285000 contains four predicted transmembrane domains. The localization analysis of the CRMP complex in tachyzoites suggests that only a small fraction is present at the plasma membrane for a short time window during invasion. Thus, experimental conditions to identify potential interaction partners of the CRMPs during invasion were modified. We used RH-CRMPA-TurboID and RH-CRMPB-TurboID and designed an invasion-specific biotinylation assay. For this, each strain was incubated on an HFF monolayer either in invasion buffer with biotin or in ENDO buffer blocking invasion with biotin for 30 min. Subsequently, mass spectrometry of biotinylated proteins was performed. No protein was significantly enriched in the invasion sample compared to the control, for both CRMPs analyzed individually and combined (S2 Data, dataset 3).

### TSP1 and MIC15 are required for host cell invasion

We performed a colocalization analysis of MIC15, TSP1, TGGT1_285500, and TGGT1_247000 by tagging them C-terminally with HALO in the RH-CRMPB-HA^floxed parasite line (Figs 5A, 5C, S5B and S5C). Additionally, we looked at HALO localization and relative signal strength 3 days post-rapamycin-induced excision of CRMPB. MIC15 and TSP1 perfectly colocalized with CRMPB (Fig 5B) and in the absence of CRMPB were partially retained within the ER (Figs 5A, 5C and S5A). In contrast, TGGT1_285500 localized to very few vesicles, which did not change in the absence of CRMPB (S5B Fig). TGGT1_247000 localized mainly to the ER, with a prominent enrichment at a basal subcompartment of the ER. Deletion of CRMPB did not result in changes of localization (S5C Fig). We also performed time lapse recordings with all HALO-tagged proteins and tagged CRMPB with HALO for reference (S1 Movie). Together, these data suggest that TGGT1_247000 and TGGT1_285500 are not part of the CRMP complex.

Next, we generated conditional mutants for these candidates, using the DiCre-system, as described above. In case of MIC15, the LoxP site was inserted into an intron, resulting in the excision of the last TSP1 domain and the transmembrane domain (S2C Fig). While deletion of TGGT1_285000 resulted in the formation of slightly smaller plaques when compared to non-induced parasites, the number of plaques was the same. Further analysis indicated a delay in replication and defects in intracellular development (S5D Fig). Thus, a functional link to

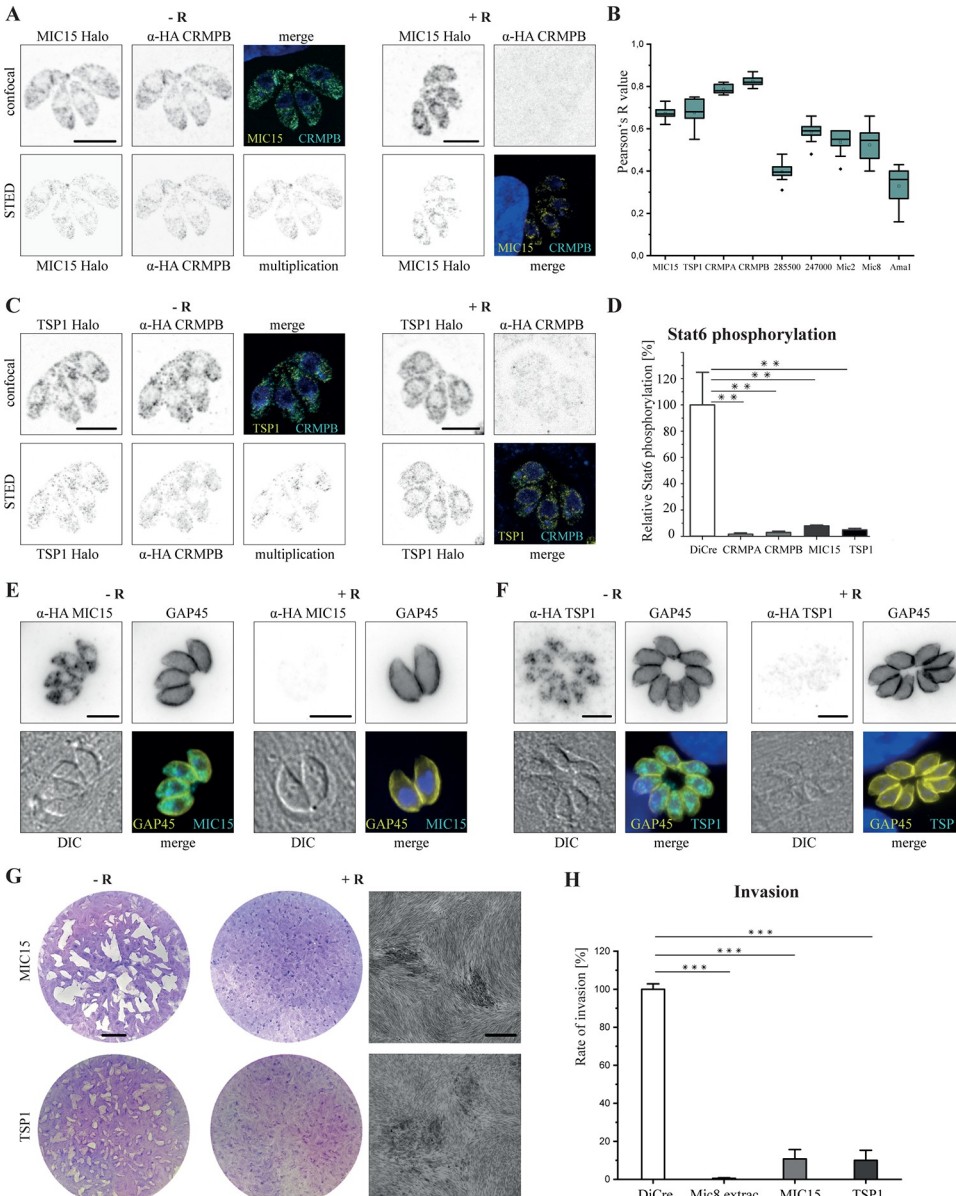

**Fig 5. Localization of potential interaction partners.** (**A**) Localization of MIC15 and CRMPB (RH-CRMPB-HA^floxed MIC15-Halo) labeled with α-HA Atto594 and Halo stained with Janelia Fluor 646. On the right localization, 72 h post-induction with rapamycin is shown. (**B**) Box blot of colocalization analysis of proteins with CRMPB 3HA, CRMPB data point is colocalization of CRMPA-3HA with CRMPB-sYFP2. Intracellular parasites were stained with α-HA Alexa 488. MIC15, TSP1, TGGT1_285500, and TGGT1_247000 were labeled prior to fixation with Janelia 646; CRMPA was labeled with α-GFP atto594; MIC2, MIC8, and AMA1 were labeled with respective antibodies and stained with Star635p. Single vacuoles of deconvolved stacks were analyzed in all planes with the color2 plugin, and the Pearson's R value of 10 vacuoles is shown (box shows 25%–75%, median indicated by line, average by little white box, outliers black dots, error bars show standard deviation). (**C**) Localization of TSP1 and CRMPB (RH-CRMPB-HA^floxed TSP1-Halo) as in (**A**). (**D**) Quantification of Stat6 phosphorylation by ROP16 as a measure of rhoptry secretion. Shown is relative Stat6 phosphorylation 72 h post-rapamycin induction from biological triplicates. (**E**) Localization of MIC15 and Gap45 in RH-MIC15-3HA^floxed induced + and − 96 h rapamycin. (**F**) Localization of TSP1 and GAP45 in RH-TSP1-3HA^floxed induced + and − 96 h rapamycin. (**G**) Plaque assay of floxed MIC15 and TSP1. Scale bar of the entire plaque assay is 5 mm; scale bar of the higher-resolution area is 500 μm. (**H**) Invasion assay of RH-MIC15-3HA^floxed and RH-TSP1-3HA^floxed 72 h post-induction. All scale bars except (**G**) are 5 μm. Statistical analysis: two-tailed Student *t* test. *** $p < 0.001$. See also S1 Data.

CRMPs seems unlikely. Similarly, deletion of tetratricopeptide protein (TGGT1_247000), although critical for parasite propagation, demonstrated a strong defect in intracellular development (S5E Fig).

In contrast, both TSP1 and MIC15 did form very small plaques, demonstrating that they are critical for parasite growth. Further analysis confirmed their critical role in rhoptry secretion, resulting in impaired host cell invasion (Fig 5D–5H). This is in good agreement of them being part of the same complex, which depends on the presence of all subunits in order to facilitate efficient host cell invasion.

In order to get more insight into how MIC15 and TSP1 associate with the CRMPs, we designed a TurboID experiment (S2 Data, dataset 4) where we wished to compare their interaction with CRMPA and CRMPB in the absence of either CRMPA or CRMPB. Therefore, we generated the parasite lines RH-CRMPB-3HA$^{floxed}$ CRMPA-TurboID and RH-CRMPA-3HA$^{floxed}$ CRMPB-TurboID. The following comparative experiments were performed. (1) Each line was grown in presence or absence of rapamycin (Fig 6A and 6C). (2) Comparing DiCre (no TurboID) to each line upon rapamycin induction (Fig 6B and 6D). When each line was compared ±rapamycin, we found that deletion of CRMPA did not result in any change for association of CRMPB with TSP1 and slight reduction of MIC15 (Fig 6A). This suggests that the interaction of CRMPB with TSP1 is independent of CRMPA. Deletion of CRMPB resulted in reduction of all components in the pulldown (CRMPA, MIC15, and TSP1; Fig 6C). Comparing + rapamycin with DiCre, expression of CRMPB-TurboID resulted in enrichment of MIC15 and TSP1 (Fig 6B). In contrast, the sole expression of CRMBA-TurboID resulted in enrichment of MIC15, but not of TSP1 (Fig 6D), indicating that CRMPA directly interacts with MIC15 and not TSP1.

Together, these results suggest that CRMPB interacts with TSP1 and that CRMPA interacts with MIC15. The stronger effect of rapamycin induction of CRMPB-3HA$^{floxed}$ is in good agreement with the IFA analysis (Fig 3D and 3E). The data suggest that CRMPA-3HA$^{floxed}$ can still interact weakly with CRMPB and MIC15 after excision of the last 2 transmembrane domains and the C-terminus following induction with rapamycin (see also S7C Fig). We did not detect the known invasion factors MIC7, MIC8, CLAMP, ND6, ND9, Ferlin 2, TGGT1_277840, TGGT1_253570 in any dataset for CRMPs-TurboID or pulldown. We did detect TgNdP2 in dataset 4 (see S2 Data), but with no difference between the +R and −R condition.

## Discussion

During invasion of the host cell, apicomplexan parasites sequentially secrete their unique secretory organelles. First, the micronemes are released and, subsequently, upon contact with the host cell, rhoptry secretion is triggered, leading to the formation of a tight interaction, the moving junction, through which the parasite invades the host cell. While some of the signaling mechanisms for microneme secretion have been described [6], the mechanisms and signaling cascades leading to rhoptry discharge are still enigmatic. While some components, such as MIC8 [8], CLAMP [10], or Nd-proteins [18], have been identified, their mode of action is in large parts unknown. Interestingly, MIC8 is only conserved in coccidia, while CLAMP homologs exist in all apicomplexans and Nd-proteins appear to be conserved throughout the super-phylum of alveolates. These findings suggest that the basic mechanism involved in rhoptry secretion is conserved, but adaptation to different lifestyles and host species required the evolution of specific invasion factors to ensure correct rhoptry discharge at the right time during invasion.

Here, we describe an invasion complex essential for rhoptry discharge that consists of both, a highly conserved core, CRMPA and CRMPB, and the more unique accessory components

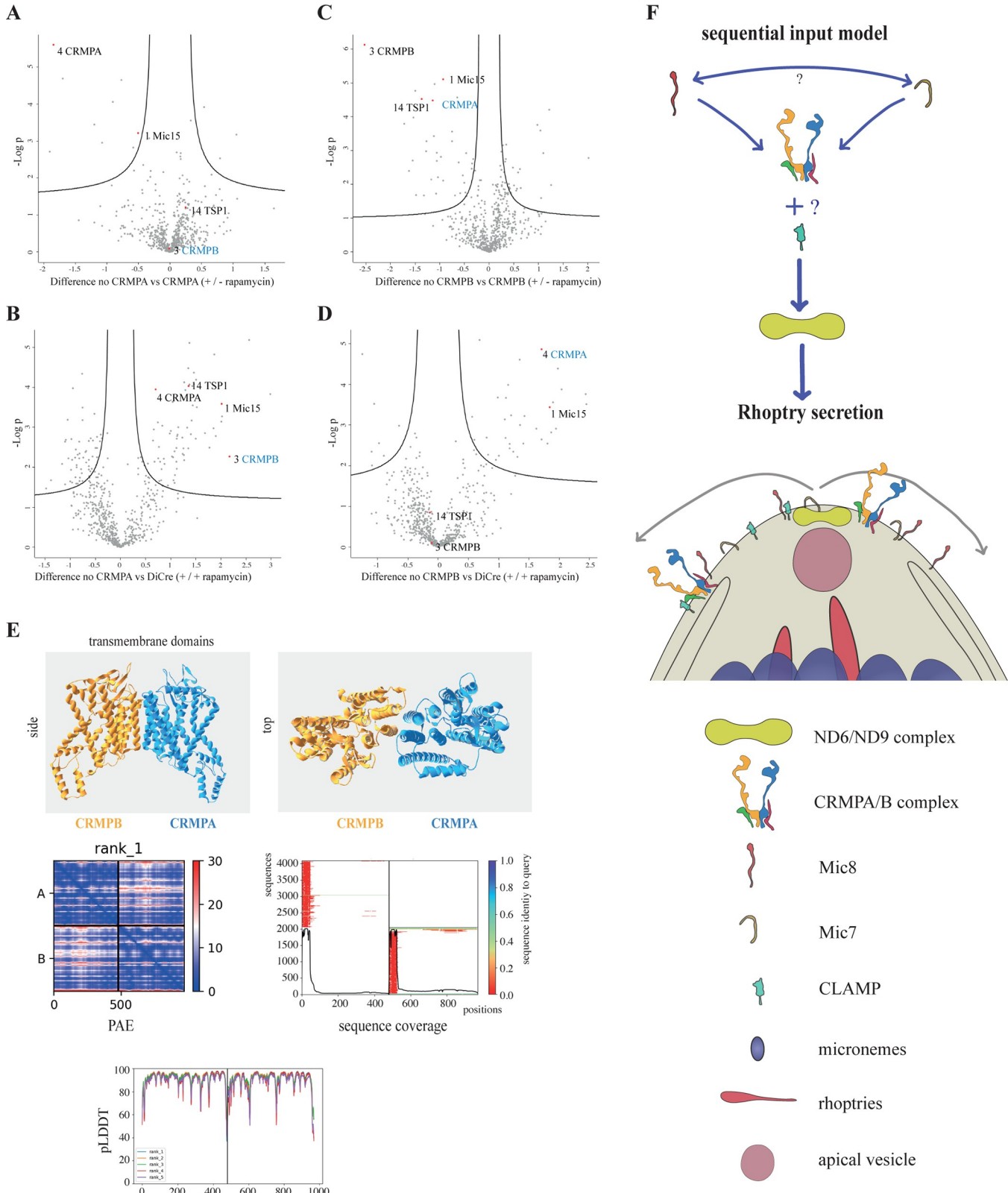

**Fig 6. CRMP complex assessment and structural modeling of CRMPs. (A–D)** Differential mass spectrometry comparing biotinylation in of RH-CRMPB-3HA^floxed CRMPA-TurboID, RH-CRMPA-3HA^floxed CRMPB-TurboID and DiCre ±72 h rapamycin treatment. The protein tagged with TurboID in the

experiment is indicated with blue writing. For changes in all other proteins, see S2 Data. (**E**) Multimer structures are predicted with Colabfold for TgCRMPA (blue) and TgCRMPB (orange). TgCRMPs form a heterodimer with their TMs, shown in side and front view; PAE score defines how sure the algorithm is to predict the relative special distance of any amino acid; the pLDDT score defines how precise local structures are predicted. Sequence alignment blots (coverage) indicate the number of sequences that could be aligned to each position. For input sequences of structure predictions, see S2 Table; for structure, see Structure 1 in S1 File. (**F**) Rhoptry secretion sequential input model. Invasion components linked to rhoptry secretion that are at or in direct proximity to the plasma membrane are highlighted. Based on the hierarchy of evolutionary conservation, we expect that the signal from the external CRMPA/B complex to the internal ND6/ND9 complex is central to rhoptry secretion, leading to downstream interaction with the apical vesicle. External species-specific receptors like MIC8 and MIC7 as well as accessory components of CRMPA/B are expected to modulate the CRMPs. As no direct interaction has been shown yet, potential mediators are unknown. In contrast to ND6/ND9 and other internal components, all secreted components are only transiently present at the apical part of the plasma membrane.

MIC15 and TSP1. In our experiments, functional disruption of either CRMPs, MIC15, or TSP1 led to a significant block in invasion, but not gliding motility. Similar to previously described rhoptry secretion factors [8,10,18], deletion of the CRMPA or CRMPB resulted in a block of rhoptry secretion.

While the CRMPs are present in all apicomplexa, the copy numbers are not identical and general sequence conservation is low. It is striking that in *Chromera velia* and even *Paramecium tetraaurelia*, a highly expanded repertoire of CRMPs can be found in the genome. It is not clear if this difference in numbers of CRMPs compared to apicomplexa results from copy number expansion in free living species or gene loss in early apicomplexa. Many apicomplexa, including coccidia, have only two members of CRMPs. In *Plasmodium* species as well as *Babesia* and *Theileria*, we could identify four members of CRMPs. In *P. berghei* and *P. falciparum*, these are found on different chromosomes; in the genome of *Babesia* and *Theileria* species, gene duplication is still visible within the genome. It seems likely that the more diverse protein family plays a role in various biological processes related to environmental sensing in *C. velia* and *P. tetraaurelia* and only one or two members specialized into their essential role for rhoptry secretion within apicomplexa. In *Cryptosporidium*, two CRMPs are also present in the genome; however, these are far more variable compared to other apicomplexa.

Our data demonstrate that the CRMPs complex is secreted to the plasma membrane during invasion. Low expression and a low secretion rate did not permit to localize its exact position at the plasma membrane during invasion, but an accumulation at the posterior pole in moving tachyzoites and post-invasion could be detected. While one would expect that a surface receptor required for rhoptry secretion is localized at the apical tip where the sensing of a suitable host cell would result in productive invasion, we only detect a relatively weak accumulation of the CRMP complex at the tip. As no signaling pathway has been identified and no drug found that triggers rhoptry secretion, one could speculate that rhoptry secretion might be triggered by direct protein–protein interactions. In contrast to classical micronemal proteins, the C-terminal tails of CRMPB and CRMPA are much longer (565 and 478 amino acids, respectively) and could act as platforms for protein–protein interactions leading to rhoptry secretion. However, our proteomic approaches did not reveal any clear candidates. The high abundance of differentially phosphorylated amino acids at the C-terminus (24 for CRMPB, 10 for CRMPA) [45] suggests that protein–protein interaction could already be primed in extracellular tachyzoites. In the RH5, RIPR, and CyRPA complex, two micronemal and a rhoptry protein (RH5) form a complex prior to rhoptry secretion [22,46]. Similar of what we expect for the CRMPs complex within Apicomplexa, only parts of this complex are conserved in all *Plasmodium* species [24].

Another open question is how CRMP integrates with the function of other essential rhoptry secretion factors, such as MIC8 and CLAMP. While CLAMP could trigger the last step in tight junction formation preceding RON2-AMA1 interaction, MIC8 seems to be part of the same step in rhoptry secretion as the CRMP complex, without any evidence for direct interaction

between both. Similarly, all attempts to identify downstream signaling molecules that interact with the complex were unsuccessful. If these interactions are as transient as the speed of the entire invasion process suggests, it will be very challenging to identify them via standard proteomic approaches, and, therefore, transient interactions with CLAMP or Nd proteins cannot be ruled out. Potentially, a mutant that allows interaction but blocks invasion could be helpful, if signaling persists in this case.

In addition to the core complex consisting of CRMPA and CRMPB, we identified MIC15 and TSP1 as additional components that are both critical for invasion. The closest ortholog of MIC15 in *Plasmodium* species is the thrombospondin-related protein (PBANKA_0707900), which seems to have a similar function in invasion [47]. It shares a TSP1 domain with TgTSP1, but sequence conservation is very low.

Deletions of CRMPA, CRMBPB, MIC15, and TSP1 result in the same phenotype; expression and localization of CRMPA depend on CRMPB, and both pulldown and TurboID experiments demonstrated a direct interaction. The last TurboID dataset in absence of either CRMPA or CRMPB (Fig 6A–6D) suggests that MIC15 interacts with CRMPA and that TSP1 does interact with CRMPB. Thus, absence of CRMPB will result in mistrafficking of TSP1 and CRMPA (Fig 3E), which will subsequently affect MIC15. Absence of CRMPA will mainly affect MIC15. Since the strength of the phenotype of CRMPA and CRMPB are identical and more severe than MIC15 and TSP1, it is likely that in addition to the proposed function at the surface, CRMPA and CRMPB act as escorter for other components of the complex. No crystal structure of any homolog of the CRMPs has been experimentally solved so far. In the recently published Alpha-Fold Protein structure database [48,49], predictions are limited by protein size; only a few structures like that of *P. falciparum* CRMP2, DICDI (*Dictyostelium discoideum*), Q86I19, and two additional DICDI homologs are partially predicted. While the extracellular part of the proteins is structurally different, the 9 transmembrane domains and some directly associated amino acids resolve quite similar in the predicted structures. We used colabfold, a google colab notebook using the AlphaFold algorithm [50] to predict the core region of CRMPB and CRMPA, as well as the multimer structure with both TMs [51]. Both transmembrane domains were predicted to form a heterodimer with each other with a very low intercomplex predicted alignment error (PAE) and a high confidence measure (pLDDT) (Fig 6E).

Both TMs folded very similar to PfCRMP2 as well as each other and assembled as an almost symmetric heterodimer. Control runs of homodimers of CRMPA with CRMPA or CRMPB with CRMPB did not result in an interaction site (S7A Fig). We also modeled the TMs of PbCRMP1-4. Only PbCRMP1 with PbCRMP3 and PbCRMP2 with PbCRMP4 resulted in a putative heterodimer structure. (S7B Fig).

MIC15 is a typical type 1 single pass transmembrane domain protein with a relatively short cytosolic tail (179 aa) as most classical micronemal proteins. It contains a long internal repeat and three extracellular thrombospondin type 1 domains. TSP1 contains a single TSP1 repeat and a potential transmembrane domain near the C-terminus. MIC15 is distantly related to TRP1 of *P. berghei*, which has been linked to oocyst egress and salivary gland invasion of sporozoites [47]. In an independent study, it was shown that MIC15 is required for rhoptry discharge and can be partially complemented by its paralog MIC14 [52].

Colabfold suggests that the most likely interaction site of both CRMPs are the TMs, as is also suggested by the interdependence expression experiment in Fig 3 as well as the weaker effect of CRMPA induction by rapamycin in the last TurboID dataset in conjunction with residual heterodimer formation after excision of the last two transmembrane domains (Figs 6A–6D and S7C). We also predicted a specific interaction partner of CRMPA clade and CRMPB clade heterodimer formation in *P. berghei*, whereas this was not the case for *B. bovis*, where most of the TMs of both members of the CRMPA clade are identical and thus

interchangeable. We fitted all the structures of the CRMP TMs individually on our TgCRMPs multimer structure. All structure predictions from the CRMPA clade reliably mapped to TgCRMPA, and all B clade to CRMPB.

It seems feasible that structure prediction could be used to generate an intracellular *Toxoplasma* interaction network as has already been done for yeast [53]. A subset analysis around known invasion factors might help to identify putative interaction surfaces that can then be tested experimentally. Multimer structure prediction could also be used to find putative interaction partners on the host cell surface, currently complicated by the fact that protein modifications like glycation is not predicted and might be essential for interaction (the same is true for phosphorylation of the C-terminus). Final confirmation of the CRMPs complex, and especially how MIC15 and TSP1 interact, might be achieved by a cryoEM structure in the future.

The most interesting questions are, however, still open. What do the extracellular parts of the CRMPs bind to? With what does the C-terminus interact, and, most importantly, how is rhoptry secretion dependent and triggered by at least three seemingly independent factors?

How can we functionally integrate previously known components of rhoptry secretion with the CRMP complex? While at this point highly speculative, based on their conservation throughout evolution, from being conserved in alveolate (Nd-proteins and CRMPs), in apicomplexa (CLAMP) down to being specific for a family (MIC8), we favor a model, where the highly conserved Nd-proteins represent a central platform that is triggered by the action of the CRMP-core complex in all alveolates. From there on, adaptations lead to the specialization for invasion, including CLAMP. Species-specific unique proteins like MIC8 or RH5 and accessory CRMP components like MIC15 and TSP1 are required for species-specific activation. Consequently, we would favor a model for the sequential action of the rhoptry secretion factors, where (in the case of *T. gondii*) MIC8 acts upstream of CRMPs, which, in turn, activate the Nd complex (Fig 6B). A recent independently performed study reached similar conclusions [54]. Here, the same four members of the CRMP complex were identified, shown to be secreted and the same phenotype using the mAID system was observed. The CRMPs also colocalize with the Nd complex at the extruded conoid. Colocalization and the identical phenotype and absence of proximity labeling suggest that they might be interlinked by an unknown mediator.

In summary, we present here a new essential invasion complex, where the CRMPs are the core components that associate with accessory proteins. Removal of a single CRMP results in functional disruption of this invasion complex. The main internal interaction interface seems to be the 9 transmembrane domains of each CRMP, resulting in heterodimer formation, which is required for the stability of the complex.

## Methods

### Sequence analysis and phylogenetic tree generation

Sequences were collected using BLAST of the CRMP1-4 of *P. berghei* (https://blast.ncbi.nlm.nih.gov/). Sequence alignment was generated with wasabi using the PRANK algorithm (https://www.ebi.ac.uk/goldman-srv/webprank/) [53,55]. Sequence analysis and phylogenetic tree generation was performed with MegaX [56]. Local alignment model (S1C Fig) was generated with geneious prime 2021.1.1, and domain models were generated in Adobe Illustrator with assistance of EMBL smart (http://smart.embl-heidelberg.de/) [36] and transmembrane domain prediction (http://www.cbs.dtu.dk/services/TMHMM/) [57].

### Cloning of DNA constructs

For C-terminal tagging, we generated a small library of tags that contain the same linker sequence GCTAAAATTGGAAGTGGAGGA (coding for AKIGSGGR), followed by a STOP

codon and a LoxP site (**TAA**ATAACTTCGTATAGCATACATTATACGAAGTTAT). Several tags were cloned into pGem or Puc19 using standard methods to allow multiple tagging with a single primer pair. Tagging sequences for SYP2, TurboID, and SNAP were a kind gift from Dorus Dadella (Addgene_22878), from Alice Ting (Addgene_107169), and from Won Do Heo (Addgene_58370) [44,58,59]. All primers used in this study were ordered from Thermo Fischer Scientific or Integrated DNA Technologies (IDT) (see S2 Table).

## Culture of *T. gondii*

Parasite culture (*T. gondii* tachyzoites RH-strain) was performed onto human foreskin fibroblasts (ATCC; SCRC-1041). Cells were cultured at 37˚C and 5% CO2 with DMEM (Sigma, D6546), supplemented with 10% FBS (BioSell FBS.US.0500), 4 mM L-Glutamate (Sigma, G7513), and 20 μg/ml gentamicin (Sigma G1397).

## Generation of transgenic parasites

Genetic modification of all parasites used in this study was performed using CRISPR/Cas9-guided double-strand break in the RH Ku80 DiCre strain [37]. For the gRNA design, EuPaGDT (http://grna.ctegd.uga.edu/) was used to select a gRNA sequence [60]. Cutting sites downstream of the STOP were preferentially used rather that recodonizing the coding region to protect the repair template from gRNA binding. Downstream cutting sites resulted in the removal of the sequence between the STOP to the cut site after repair. The gRNA sequence was cloned into and transiently expressed from a Cas9-YFP vector [61]. All tags used in the lab are flanked by the same linker sequence upstream and by a STOP-LoxP sequence downstream. On both sides, 50 bp of homology was added via PCR to generate the repair templates for homologous recombination using Q5 polymerase (NEB). For internal tagging, the reverse primer was tag specific omitting the STOP-LoxP. Upstream or Intron-LoxP sequences were integrated using a 100-bp single oligo, consisting of 33 bp homology on both sides of the LoxP sequence. For transfection, 10 to 12 μg per Cas9 plasmid, purified PCR product of 200 μl reaction volume and 2 to 5 μl of LoxP repair oligo (100 μm) were pooled and purified using a PCR purification Kit (Blirt). Freshly lysed parasites were transfected using the Amaxa 4D-Nucleofactor system (Lonza) P3 primary cells kit, program FI-158. Parasites were mechanically released, filtered with a 3-μm filter and sorted for YFP expression into a 96-well plate (5 and 10 events; 48 wells each) (FACSARIA III, BD Biosciences), 24 to 48 h post-transfection. Individual plaques were screened by PCR, and all modified loci were sequenced to confirm correct modification. For multiple modification, subsequent rounds of transfection, sorting, and genotyping took place. For primer sequences, gRNAs, and tags used as well as parasites generated, see S2 Table and S3 Data.

## Labeling and pulldown

For dataset 1 in S2 Data, direct pulldown, parasites (RH-CRMPA-3HA, RH-CRMPB-3HA, and RH-DiCre) were mechanically released from host cells, filtered and washed with PBS, then stored at −80˚C. For dataset 2 in S2 Data, biotin labeling, parasites (RH-CRMPB-TurboID or RH-DiCre) were mechanically released and filtered, then parasites were incubated at 37˚C for 30 min in 1 ml of DMEM media. Samples were centrifuged at 1,500*g* for 5 min and washed with PBS (0˚C) three times, then stored at −80˚C. Dataset 2 in S2 Data is not shown in the figure as the more comprehensive dataset 4 in S2 Data (just–R) is shown in Fig 4C and 4E. Results are very similar; full data can be compared in S2 Data. For dataset 3 in S2 Data, parasites (RH-CRMPA-TurboID or RH-CRMPB-TurboID) were resuspended in 2 ml of ENDO buffer with 150 μM of Biotin (C) or in HBSS (with 25 mM HEPES, 1% FCS, and 150 μM

Biotin(R)) on a 6-cm dish of HFF and incubated at 37˚C for 30 min. After incubation, the dish was scratched and centrifuged in 50 ml of ice-cold PBS at 1,500$g$ for 10 min at 0˚C. Supernatant was discarded, and 2 identical washes performed followed by resuspension in 1 ml of PBS and a final spin after transfer into a 1.5-ml tube. Sample was stored at −80˚C. For dataset 4 in S2 Data, RH-CRMPB-3HA$^{floxed}$ CRMPA-TurboID, RH-CRMPA-3HA$^{floxed}$ CRMPB-TurboID, and RH-DiCre were induced with rapamycin or treated with DMSO and cultured for 72 h. Then, biotin labeling was performed as for dataset 2 in S2 Data. Parts of dataset 4 in S2 Data are shown in Fig 4C and 4E (−rapamycin), the rest in Fig 6. Raw data in S2 Data.

For HA pulldown, parasite pellet was lysed for 30 min on ice in 1 ml of RIPA buffer (0.5% sodium deoxycholate, 150 mM NaCl, 1 mM EDTA, 0.1% SDS, 50 mM Tris–HCL (pH 8.0), 1% Triton-X-100 with 1:100 protease and phosphatase inhibitor cocktail, Sigma). Now, 50 μL (0.25 mg) of Pierce α-HA magnetic beads and 350 μL of 0.05% TBS-T were vortexed gently. Beads were placed into a magnetic stand to collect the beads against the side of the tube, and supernatant was removed and washed with 1 ml of TBS-T for 1 min. Now, the lysed sample was added (spin and only use supernatant) and incubated for 30 min with gentle mixing. Beads were washed with 300 μl of TBS-T, then 300 μl of ultrapure water.

For TurboID pulldown, the pellet was lysed for 30 min on ice in 1 ml of RIPA buffer with 1% Triton-X-100 and 1:100 protease inhibitor. Per sample 50 μl (100 μl for the invasion specific second experiment) of beads (Dynabeads MyOne Streptavidin T1, Invitrogen) were washed 3 times with 1 ml of PBS using the magnet system. Beads were responded with cleared lysed parasites in RIPA buffer and incubated 1 h at RT while gently mixing. Beads were washed 5 times with 1 ml RIPA buffer without Triton. Beads were washed three times with 50 mM NH$_4$HCO$_3$, and 10% of the volume was separated for western blot controls.

## Mass spectrometry

Beads were incubated with 10 ng/μL trypsin in 1 M urea 50 mM NH$_4$HCO$_3$ for 30 min, washed with 50 mM NH$_4$HCO$_3$, and the supernatant digested overnight (ON) in presence of 1 mM DTT. Digested peptides were alkylated and desalted prior to LC–MS analysis.

For LC–MS/MS purposes, desalted peptides were injected in an Ultimate 3000 RSLCnano system (Thermo), separated in a 15-cm analytical column (75 μm ID with ReproSil-Pur C18-AQ 2.4 μm from Dr. Maisch) with a 50-min gradient from 4% to 40% acetonitrile in 0.1% formic acid. The effluent from the HPLC was directly electrosprayed into an Orbitrap Exploris 480 (Thermo) operated in data-dependent mode to automatically switch between full scan MS and MS/MS acquisition. Survey full scan MS spectra (from m/z 350 to 1,200) were acquired with resolution R = 60,000 at m/z 400 (AGC target of $3 \times 10^6$). The 20 most intense peptide ions with charge states between 2 and 5 were sequentially isolated to a target value of $1 \times 10^5$ and fragmented at 30% normalized collision energy. Typical mass spectrometric conditions were as follows: spray voltage, 1.5 kV; heated capillary temperature, 275˚C; ion selection threshold, 33.000 counts.

MaxQuant 2.0.1.0 was used to identify proteins and quantify by iBAQ with the following parameters: Database UP000005641_Toxoplasmagondii_20201123.fasta; MS tol, 10 ppm; MS/MS tol, 20 ppm Da; Peptide FDR, 0.1; Protein FDR, 0.01 min; Peptide Length, 7; Variable modifications, Oxidation (M); Fixed modifications, Carbamidomethyl (C); Peptides for protein quantitation, razor and unique; Min. peptides, 1; Min. ratio count, 2. Identified proteins were considered as interaction partners of the bait if their MaxQuant iBAQ Z-score normalized values were as follows: $\log_2$(CRMPA-3HA) − $\log_2$(WT) + $\text{Log}_2$(CRMPB-3HA) − $\log_2$(WT) > 1.8. The TurboID datasets were blotted in a volcano blot. Proteins that have a known function, are not expressed, and have no localization data by lopit or a phenotypic

score $> -1$ were excluded from further analysis. Proteins with nuclear, proteasomal, mito-chondrial, rhoptry, and apicoplast localization were also excluded. For display and analysis, the Perseus software [62] was used. Data have been uploaded to the PRIDE repository [63] PXD035654 (datasets 1–3 in S2 Data), PXD031649 (dataset 4 in S2 Data).

### Micronemal secretion assay

Secretion of micronemes was assessed essentially as published before [64]. Parasites were treated for 72 h with ±50 nM rapamycin and then mechanically released and filtered, centri-fuged for 5 min at 4˚C at 1,500$g$ and washed with cold intracellular buffer (5 mM NaCl, 142 mM KCl, 1 mM MgCl$_2$, 2 mM EGTA, 5.6 mM glucose, 25 mM HEPES (pH 7.2)), and counted. Per condition $6 \times 10^7$ parasites were then resuspended in 100 µl intracellular buffer with 2 µM Ci A23187 or DMSO and incubated for 30 min at 37˚C. Samples were centrifuged for 5 min at 4˚C at 1,500$g$, supernatant collected and cleared by a second centrifugation step, and stored at −80˚C.

For western blotting, 10 µl of the supernatant was denatured together with 4 µl of 4× load-ing dye and 1.6 µl 1 M DTT and loaded on a 4% to 20% precast gel (BioRad, 4561096). Anti-bodies against MIC2 and GRA1 were used to image the membrane using the Odyssey CLX-1849 (LI-COR). Blots were quantified and relative MIC2/GRA1 signal is shown for 3 (RH-CRMPA-3HA$^{floxed}$ and RH-CRMPB-3HA$^{floxed}$) or 2 (DiCre) biological replica.

### Microscopy assays

All microscopy was performed at a Leica DMi8 widefield microscope with a DF C9000 GTC camera or at the Abberior 3D STED microscope (STED microscopy). For live cell microscopy, a heated chamber and 5% CO2 was used.

### Immunofluorescence labeling

Parasites stained with Halo or SNAP were labeled prior to fixation. SNAP was labeled with SNAP-Cell 647-SiR (from NEB Biolabs), Halo was labeled with Janelia Fluor 646 Halo ligand (Promega). Dyes were stained life for 30 min to 2 h, washed 3 times with PBS, and incubated for 15 min in media. Cells were fixed 15 min with 4% PFA in PBS, washed 3 times with PBS, and blocked for 30 min with 2% to 3% BSA. If the sample was permeabilized, 0.2% Triton was added here. Primary antibody incubation was performed at RT for 1 h in blocking buffer, fol-lowed by 3 washes with PBS. The sYFP2 tags were stained with rabbit polyclonal α–GFP (Abcam, #290). The secondary antibody was incubated together with Hoechst if desired in blocking buffer at RT for 1 h, followed by 5 washes with PBS. Samples were imaged directly in PBS or mounted with Prolong Gold, Prolong Diamond, or Prolong Glass (Thermo Fischer Sci-entific). Secondary antibodies used were Alexa 488, Atto 594 (Thermo Fischer Scientific), Abberior Star 580, Abberior 635P, and Abberior StarRed (Abberior). Staining of Biotin-labeled TurboID samples was performed with directly labeled Streptavidin (Alexa488 or Alexa594) (Thermo Fischer Scientific).

### Plaque assay

Plaque assays were performed on confluent HFFs in 6-well plates in triplicates. Of each para-site line, 1,000 parasites were used per well, incubated for 7 days with 4 ml of media per well ±50 nM rapamycin as previously described [65]. A RH-DiCre and RH-MIC8extrac.$^{floxed}$ con-trol was run in parallel. Overview images were acquired with a cell phone camera against an

illuminated white back ground. Inserts were generated with the Leica DMi8 widefield micro-
scope using a $10 \times 10$ field stitched image with 10% overlay.

### Gliding assay

Parasites were induced with ±50 nM rapamycin 3 days prior to gliding assay. Live-cell micros-
copy chambers (μ-Slide 8 well (Ibidi)) were incubated with 50% FBS at RT for 1 h. Freshly fil-
tered parasites were incubated in HBSS with 1% FBS and 25 mM HEPES and directly
transferred into the μ-Slide 8 well and imaged in the preheated Leica DMi8 microscope with 1
fps with a 20× air objective.

### Invasion assay

Fixed invasion assay (red/green assay) was performed on HFFs cells on glass slides in 24-well
plates. Parasites after ±rapamycin were mechanically released and filtered and adjusted to
$3 \times 10^6$ parasites in 200 μl of DMEM. Parasites were settled for 10 min at RT and incubated at
37˚C for 20 min, followed by fixation of 4% PFA for 15 min. Sample was washed 3× with PBS,
blocked without Triton (3% BSA in PBS) for 1 h, stained with anti-Sag1 antibody [66], washed
3× with PBS, and stained with anti-mouse Alexa-488. Sample was briefly fixed with PFA,
washed 3× with PBS, blocked with Triton (3% BSA in PBS, 0.2% Triton-X-100) for 1 h, stained
with α-Gap45 antibody, washed 3× with PBS, and stained with anti-rabbit Abberior-580.

For 24-h invasion/replication assay, $5 \times 10^6$ of freshly release and filtered tachyzoites were
allowed to invade for 20 min. Treatment ±rapamycin was started 72 h before invasion. After
24 h, cells were fixed and stained with α-Gap45 and α-HA (Roche, 3F10, monoclonal). In the
+R sample, no intracellular parasites were observed that expressed detectable HA. Relative par-
asite number within vacuoles was counted for three biological replica.

Live cell imaging of invasion was performed in HBSS, 25 mM HEPES (pH 7.3), 1% FCS at
the Leica DMi8. Parasites were mechanically released, filtered, resuspended in prewarmed
medium, and directly imaged for 20 min in a heated microscope chamber with 5% CO2.

### Evacuole assay

Freshly egressed and filtered tachyzoites 72 h after incubated with ±rapamycin were adjusted
to $5 \times 10^6$ parasites in 250 μl and incubated with 1 μM of Cytochalasin D (CytoD) for 10 min
at RT. Parasites were added to confluent HFF on glass slides in 24-well plates and incubated
for 15 min at 37˚C. Cells were fixed and blocked in 3% BSA in PBS. Samples were stained with
anti-ROP1 mAb T52A3 (Soldati-Favre lab) and anti-Sag1 (Abcam 138698).

### Rhoptry secretion assay using ROP16 mediated STAT6 phosphorylation

Rhoptry secretion was assessed essentially as described [67]. Parasites were treated for 72 h
with ±50 nM rapamycin and then mechanically released and filtered, centrifuged for 5 min at
4˚C at 1,500*g*, and washed with full medium. Per replica and condition 250 μl of parasites were
added on a 24-well plate with confluent HFF on glass slides, allowed to settle for 20 min at RT,
and then incubated for 20 min at 37˚C. Cells were fixed with ice-cold methanol at −20˚C for 8
min and stained with anti-STAT6-P (1:600, (Cell Signaling 56554S)) and anti-Sag1 antibody
[66].

### STED microscopy

Super-resolution microscopy was performed at the Abberior 3D STED microscope, equipped
with 3 color STED. Imaging settings were individually adjusted to the sample to maximize

signal to noise and resolution and vary depending on signal strength. Generally, $60 \times 60 \times 250$ nm sampling was performed for confocal mode, $30 \times 30 \times 250$ nm for 2D STED and $60 \times 60 \times 60$ nm for 3D STED.

## Image analysis

Deconvolution of widefield images was performed with Huygens essentials (SVI). Microscopy images were analyzed using Fiji [68]. Colocalization analysis was performed using the Coloc 2 plugin (https://github.com/fiji/Colocalisation_Analysis). The localized colocalization display was performed by multiplication of every single pixel between both channels. This helps to visualize if the colocalization is position independent or not, no quantification is performed.

For the relative signal analysis in S5A Fig, the relative localization of MIC15 was quantified in the presence and absence of CRMPB. Apical relative signal is percentage of signal apical of the nucleus (area selected manually) of total signal within one tachyzoite. Circles represent individual tachyzoites; the center plane of a deconvolved stack was analyzed. In the post-ER analysis, the nucleus was automatically thresholded using the Huang algorithm of the Hoechst staining (this generates a threshold surrounding the nucleus) and this threshold was applied onto the Halo signal, giving rise to the ER signal. The remaining fluorescence intensity within the parasite is defined as the post-ER signal. Each dot represents all planes of a vacuole of a deconvolved stack. In all analysis, sample was normalized with a background control.

## Structure prediction

Multimer structure prediction was performed using Colabfold [49,50,51,69] (https://github.com/sokrypton/ColabFold). We used the mmseqs2 notebooks, both v.12 and v.13 (v. 1.3 always runs 5 models resulting in a higher failure rate to finish the prediction before timeout). The transmembrane regions of CRMPA, CRMPB, and MIC15 were predicted alone and in combination, as a control also CRMPB with CRMPB and CRMPA with CRMPA as dimer prediction. Same was performed for the sequences upstream and downstream of the TMs. Additionally, single sequences were submitted to Robetta (using Rosettafold webserver: https://robetta.bakerlab.org/) [70]. To define the TMs of the CRMPs, we used the pdb file of PfCRMP2 from the alphafold database and mapped the amino acids to the alignment to define the TMs for all other CRMPs. Only for CRMPB this had to be manually adjusted due to bad alignment quality upstream of the TMs. For all input sequences, see S2 Table; structures are in S1 File.

## Data analysis

Graphs are generated with MS Exel or Origin pro (OriginLab). All other image analysis was performed using Fiji. Figures were generated with Adobe Illustrator.

## Supporting information

**S1 Movie. Intracellular imaging.** Endogenous tagging of CRMPA in the extracellular part (RH-Halo-CRMPA-3HA[floxed]) or C-terminal tagging with Halo (all other proteins) shows that they are very dynamic during intracellular development.
(MP4)

**S2 Movie. Localization of CRMPA and CRMPB during invasion.** Invasion of Halo labeled (Janelia 646) CRMPA and CRMPB into HFF cells (RH-Halo-CRMPA-3HA[floxed] RH-CRMPB-Halo). Note how part of the protein is visible at the basal end of the parasite, both prior to invasion and after. In one event, signal is observed at the tight junction.
(MP4)

**S1 Data. All numerical input data for graphs shown in figures.**
(XLSX)

**S2 Data. Pulldown and TurboID mass spectrometry data.**
(XLSX)

**S3 Data. DNA oligonucleotides used in this study.**
(XLSX)

**S1 Table. Sequences of all CRMPs and Alignment for Fig 1A.**
(TXT)

**S2 Table. List of gRNAs used, parasites generated, and input sequences for structure predictions.**
(DOCX)

**S3 Table. Sequences of all CRMPs and Alignment for S1A Fig.**
(TXT)

**S1 Fig. Phylogenetic analysis and Lopit map placement.** (**A**) Phylogenetic relationship of CRPMs is shown after alignment of the conserved core region using the PRANK algorithm; bootstrap analysis (100×) supports most branches. Compare with Fig 1A. See also S3 Table. (**B**) Spatial proteomic map of all *T. gondii* proteins [30], the micronemal cluster (MIC) and the ER1 cluster are indicated. Insert on the right below the ER cluster shows a separate "minicluster" of CRMPB and CRMPA, grouping together with Thrombospondin typ 1 domain-containing protein (TSP1; TGGT1_277910) and MIC15 (TGGT1_247195). See also https://proteome.shinyapps.io/toxolopittzex/ [30]. (**C**) A local conservation map and local protein identity and conservation. Insertion are indicated by gray boxes.
(TIF)

**S2 Fig. Gene tagging strategies.** (**A**) Shown is an overview of the domain architecture and the gene CRMPB of *T. gondii*. Arrowheads indicate the position of the C-terminal (including the C-terminal LoxP site) and extracellular tag in the gene and the protein model. Position of the LoxP in the intron and the local sequence is shown; relative position within the domain model is indicated by a line and lost sequence after rapamycin induction is indicated by a red bar. (**B**) Overview of the domain architecture and the gene of CRMPA. Note that only the last transmembrane domain and the C-terminus is lost upon induction with rapamycin. (**C**) Overview of the domain architecture and the gene MIC15. Position of the LoxP within the intron and the part of the gene lost after rapamycin induction is indicated. (**D**) Overview of domain architecture of MIC8. The upstream LoxP sequence is 6 bp upstream of the start site (position indicated by an arrow head); the internal LoxP site is 3 bp after the last EGF-like domain inserted as coding as indicated, inserting the protein sequence ITSYSIHYTKLS into the protein (coding sequence in lower case, protein sequence in green). Shielding mutations inserted are indicated with orange letters. Several domains are indicated: CLECT (c-type Lectin or carbohydrate recognition domain), ERC (Ephrin-receptor like), EGF-like (Epidermal growth factor-like), TM (transmembrane domain), Kringle (Kringle domain), EGF (Epidermal growth factor), coil coil (alpha helical coil coil domains), TSP1 (thrombospondin type-1 repeat). (**E**) Genotyping of RH-CRMPA-3HA^floxed and RH-CRMPB-3HA^floxed after 72 h induction with rapamycin. The respective primers bind outside of the floxed region and are indicated in (**A**) and (**B**) with arrows. Estimated sizes are indicated below in basepairs.
(TIF)

**S3 Fig. The impact of CRMPs on micronemes and colocalization.** (**A**) Quantification of microneme secretion measured by secretion of MIC2 relative to GRA1 secretion. DiCre, RH-CRMPA-3HA^floxed, and RH-CRMPB-3HA^floxed were measured 72 h post-rapamycin treatment. See also S1 Data. The abundance and localization of (**B**) AMA1, (**C**) MIC2, and (**D**) MIC8 relative to CRMPB was analyzed in presence and absence (72 h rapamycin) of CRMPB in RH-CRMPB-3HA^floxed. (**E**) Parasites with extracellular SNAP-tagged CRMPB (RH-SNAP-CRMPB-3HA^floxed CRMPA-sYFP2) or extracellular Halo-tagged CRMPA (RH-Halo-CRMPA-3HA^floxed CRMPB-sYFP2) were transiently transfected with (**E**) ERD-GFP and (**F**) GalNac YFP (TGGT1_259530-YFP) [71] and labeled (SNAP-Cell or Halo-Janelia 646) and imaged live 48 h post-transfection. Scale bars are 5 µm.
(TIF)

**S4 Fig. Extracellular tagging of CRMPs.** (**A**) Localization of HALO (Janelia Fluor 646) within the extracellular domain of CRMPA (see S2 Fig) in respect to C-terminally tagged CRMPA (3HA) and C-terminally tagged CRMPB (sYFP2). (**B**) Localization of RH-CRMPB-3HA^floxed CRMPA-sYFP2 was stained with for HA (Star635P) and sYFP2 (Atto594). Compare with Fig 2A. (**C**) Localization of SNAP (Sir647) within the extracellular domain of CRMPB (see S2 Fig) in respect to C-terminally tagged CRMPB (3HA) and C-terminally tagged CRMPA (sYFP2). (**D**) Parasites with extracellular CRMPB SNAP (Janelia 646) after pulse invasion in the presence of proponol. The putative invasion site is marked with an arrow. Scale bars are 5 µm.
(TIF)

**S5 Fig. Characterization of potential CRMP interactors.** (**A**) The relative localization of MIC15 was quantified in the presence and absence of CRMPB. Analysis was performed manually (apical of total fluorescence) and automatically (post-ER of total fluorescence). For details, see Methods section and S1 Data. (**B**) Colocalization of TGGT1_285500 (Halo) with CRMPB and localization of 285500 in the absence of CRMPB (72 h). (**C**) Colocalization of the tetratricopeptide repeat containing protein (TGGT1_247000) (Halo) with CRMPB and localization of 247000 in the absence of CRMPB (72 h). (**D**) Localization of 3HA tagged 285500 ±rapamycin (72 h). Plaque assay after 7 days shows a mild growth phenotype. (**E**) Localization of 3HA tagged 247000 ±rapamycin (72 h). No plaques could be detected 7 days after rapamycin induction. Note that there is some crosstalk with the Gap45 signal in (C) and (D). Scale bars of the plaque assay are 5 mm (scale bar of the higher-resolution area is 500 µm); all other scale bars are 5 µm.
(TIF)

**S6 Fig. Analysis of invasion using RON2-SNAP.** In the cKO parasites of CRMPB and CRMPA, RON2 was endogenously tagged with SNAP. Shown are maximum projections after 20 min of invasion (72 h post-rapamycin induction); invaded parasites are circled. Both invaded parasites in the plus rapamycin group represent the only tachyzoite with a clear "invaded" RON2 staining pattern; minus rapamycin are representative stacks.
(TIF)

**S7 Fig. Structural modeling of CRMPs.** (**A**) Homodimer modeling for (1) CRMPA and (2) CRMPB result in overlay of proteins. The PAE scores show good relative prediction for single chains but indicate that both chains do not interact, despite the high IDDT values. Sequence alignment blots indicate the number of sequences that could be aligned to each position. (**B**) Multimer heterodimer prediction of PbCRMP1/3, PbCRMP1/4, PbCRMP2/3, and PbCRMP2/4. CRMPA clade members are colored light blue; CRMPB clade members are colored orange. PAE scores indicate that PbCRMP1/3 and PbCRMP2/4 do interact and that PbCRMP1/4 and PbCRMP2/3 do not interact. PbCRMP3 and PbCRMP4 are predicted in two fragments with

an unstructured part (indicated by the low local pLDDT score). (**C**) Prediction of heterodimer formation of CRMPA and CRMPB transmembrane domains after induction of RH-CRMPA-3HA^floxed resulting in loss of the C-terminus and the last 2 transmembrane domains. New C-terminus is indicated in red. Notice the change in PAE score compared to full-length TMs shown on the right. Input sequences are in S2 Table, Structure B-H in S1 File.
(TIF)

**S1 Raw Images.** (**A**) Raw image of DNA agarose gel shown in S2E Fig. On the right side, the negative control (genotyping the non-floxed locus) is shown (data not included in S2E Fig). (**B-D**) Raw images of western blots used for the quantification of micronemal secretion in S3A Fig. MIC2 (upper bands) and GRA1 (lower band) are probed with the same secondary antibody 72 h post-rapamycin treatment. For details, see Methods section.
(PDF)

**S1 File. The structural pdb files Structure A-H for all structures shown in Figs 6E, S7A, S7B and S7C.**
(ZIP)

## Acknowledgments

We would like to thank colleagues for sharing of primary antibodies and plasmids used in this study. We thank Sonja Härtle and Marina Kohn for access to their FACS as well as the expert support. We thank Wei Li for help with the micronemal secretion assay. Hirdesh Kumar assisted us in evaluation of the modeled protein structures.

## Author Contributions

**Conceptualization:** Mirko Singer, Markus Meissner.

**Data curation:** Mirko Singer.

**Formal analysis:** Mirko Singer, Kathrin Simon, Markus Meissner.

**Funding acquisition:** Mirko Singer, Markus Meissner.

**Investigation:** Mirko Singer, Kathrin Simon.

**Methodology:** Mirko Singer, Kathrin Simon, Ignasi Forné.

**Project administration:** Markus Meissner.

**Software:** Ignasi Forné.

**Supervision:** Mirko Singer, Ignasi Forné, Markus Meissner.

**Visualization:** Mirko Singer, Ignasi Forné.

**Writing – original draft:** Mirko Singer, Markus Meissner.

**Writing – review & editing:** Mirko Singer, Kathrin Simon, Ignasi Forné, Markus Meissner.

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
