## [Editor Report · Decision Letter 0]

7 Mar 2022

Dear Dr Singer, 

Thank you for submitting your manuscript entitled "A central protein complex essential for Invasion in Toxoplasma gondii" for consideration as a Research Article by PLOS Biology.

Your manuscript has now been evaluated by the PLOS Biology editorial staff, as well as by an academic editor with relevant expertise, and I am writing to let you know that we would like to send your submission out for external peer review.

Once your full submission is complete, your paper will undergo a series of checks in preparation for peer review. Once your manuscript has passed the checks it will be sent out for review. To provide the metadata for your submission, please Login to Editorial Manager (https://www.editorialmanager.com/pbiology) within two working days, i.e. by Mar 09 2022 11:59PM.

If your manuscript has been previously reviewed at another journal, PLOS Biology is willing to work with those reviews in order to avoid re-starting the process. Submission of the previous reviews is entirely optional and our ability to use them effectively will depend on the willingness of the previous journal to confirm the content of the reports and share the reviewer identities. Please note that we reserve the right to invite additional reviewers if we consider that additional/independent reviewers are needed, although we aim to avoid this as far as possible. In our experience, working with previous reviews does save time. 

If you would like to send previous reviewer reports to us, please email me at dummarino@plos.org to let me know, including the name of the previous journal and the manuscript ID the study was given, as well as attaching a point-by-point response to reviewers that details how you have or plan to address the reviewers' concerns. 

Given the disruptions resulting from the ongoing COVID-19 pandemic, please expect some delays in the editorial process. We apologise in advance for any inconvenience caused and will do our best to minimize impact as far as possible.

Kind regards,

Dario

Dario Ummarino, PhD

Senior Editor

PLOS Biology

dummarino@plos.org

---

## [Decision Letter · Decision Letter 1]

10 May 2022

Dear Dr Singer,

Thank you for your patience while your manuscript "A central protein complex essential for Invasion in Toxoplasma gondii" was peer-reviewed at PLOS Biology. Your manuscript has been evaluated by the PLOS Biology editors, an Academic Editor with relevant expertise, and by several independent reviewers.

As you will see in the reviewer reports, which can be found at the end of this email, although the reviewers find the work potentially interesting, they have also raised a substantial number of important concerns. Based on their specific comments and following discussion with the Academic Editor, it is clear that a substantial amount of work would be required to meet the criteria for publication in PLOS Biology. However, given our and the reviewer interest in your study, we would be open to inviting a comprehensive revision of the study that thoroughly addresses all the reviewers' comments. Given the extent of revision that would be needed, we cannot make a decision about publication until we have seen the revised manuscript and your response to the reviewers' comments. Your revised manuscript would need to be seen by the reviewers again, but please note that we would not engage them unless their main concerns have been addressed. 

IMPORTANT:

Having discussed the reviews with the Academic Editor, we think you should clearly show whether the depletion of CRMPA or CRMPB affect the secretion of microneme proteins (point #2 from reviewer 1). In addition, you should further investigate the specific interactions within the protein complex comprising CRMPA, CRMPB, Mic15, and Tsp1 (points #1 from reviewer 2 and 3). Moreover, we think that the suggestion from reviewer 1 (point 3) to provide additional mechanistic data on the signalling pathways affected by the depletion of CRMP complexes would not be crucial for the current manuscript and may be a subject of future investigations. Finally, the speculations around the implications of the structural modelling should be toned down or removed (points 4 from reviewer 1 and 2). All the other concerns raised by reviewers should be carefully and thoroughly addressed. 

We appreciate that these requests represent a great deal of extra work, and we are willing to relax our standard revision time to allow you 6 months to revise your study. Please email us (plosbiology@plos.org) if you have any questions or concerns, or envision needing a (short) extension.

**IMPORTANT - SUBMITTING YOUR REVISION**

*Resubmission Checklist*

*Published Peer Review*

*PLOS Data Policy*

*Blot and Gel Data Policy*

Sincerely,

Dario

Dario Ummarino, PhD

Senior Editor

PLOS Biology

dummarino@plos.org

REVIEWS:

Reviewer #1: In this manuscript, the authors characterized a novel protein complex in Toxoplasma parasites for rhoptry release, further impairing parasite invasion. Two cysteine repeat modular proteins (CRMPs), named CRMPA and CRMPB, are apicomplexan-specific members. The authors used the Di-Cre system to delete individual CRMP proteins and found that both are essential for parasite invasion. Further, the authors used pull-down and BioID assays to determine the interacting partners with both CRMPs. Two proteins, Mic15 and TSP1, were enriched as the top candidates interacting with both CRMPs. The manuscript took multiple strategies to discover and validate protein-protein interactions and the subcellular localization of the proteins of interest, which enhances the rigor of this work. In addition, the generation of multiple knockouts for the components within this essential protein complex strengthens the conclusion. Overall, this manuscript studied a fundamental question of how the apicomplexan parasites shed their invasion factors for their infection. The authors provided compelling data from microscopy and phenotypic assays for proving their hypothesis. I have a few comments below.

1. The authors identified a novel protein complex comprising CRMPA, CRMPB, Mic15, and Tsp1. I wonder if Mic15 and Tsp1 interact with CRMPA or CPMPB directly or they associate with CRMPA via CPMPB and vice versa? A biotinylation assay in the strains lacking CRMPA or CRMPB may answer this question. 

2. Does the depletion of CRMPA or CRMPB affect the secretion of microneme proteins? It would be a good control for distinguishing the roles of microneme and rhoptry release in parasite invasion. 

3. The authors mentioned several previously reported determinants for rhoptry release. It would be helpful to dissect if the depletion of CRMP complexes will not affect other pathways. 

4. Although the authors predicted the structures of CRMPs using the latest algorithm to build the 3-D model of the CRMP complexes, I feel the authors may need to keep cautious to use this model for speculation and prediction. It would be great to move this part from the results to the discussion section. 

Reviewer #2: This manuscript present new findings on two Cysteine Repeat Modular Proteins in T. gondii. CRMPA and CRMPB are homologous of proteins in Plasmodium, where they have been shown to be important in the invasion by sporozoites, but interestingly not merozoites. These orthologs were also predicted by be essential in a genome wide CRISPR screen in T. gondii. Hence the investigators take the rather predictable approach of deleting them using an inducible DiCre system to test their function. Not unsurprisingly, they are found to be important. The key new finding is that these proteins appear to be necessary for rhoptry secretion. Additionally, there is evidence that they interact with microneme proteins in a complex, which is needed for invasion. What is less clear is why the distribution of the proteins (the secretory system in general) does not match the putative function (apical complex). It is also unclear why the authors propose a complex model for the extracellular domains of CMRPA/CRMPB that is not well supported by data, while more direct alternatives are not acknowledged.

Specific comments

1) The abstract is unnecessary complicated by negative comments about prior work on individual microneme proteins. Not expectantly, many microneme proteins have overlapping or redundant roles and hence knockouts often do not reveal strong or "essential phenotypes". However, this is not the point of the present manuscript, which rather focuses on a new complex important for rhoptry secretion and invasion. I would suggest focusing on the positive data for the new findings rather than disparaging previous work for perceived deficiencies that are entirely consistent with the biology.

2) The localization data for CRMP1 and CRMPB is hard to reconcile with the proposed function in ROP secretion. The CRMP proteins appear to be widely distributed in the secretory pathway, but only transiently or weakly associated with the apical tip. The use of an N terminal tag is used to show that CRMPs are secreted to the surface, but these experiments are complicated by the fact that parasites are imaged after invasion, leaving a lot of time for redistribution (Fig 2C). Additionally, the parasites being imaged appear to express YFP, and hence it is not clear that the detection of GFP added prior to invasion is not simply due to cross reaction. Many antibodies to GFP cross-react to YFP and it is not clear what is the specificity of the reagents used here? Regardless, it would more informative to visualize the extracellular tag in parasites that are free of host cells, for example when stimulated to secrete micronemes and when the conoid is extended. Is there evidence for external display of the protein at the apical end in such parasites? 

3) Although the authors favor a view that CRMPs are needed for ROP secretion, their primary localization does not match this function but rather appears to be at the posterior end (after invasion). As well, they appear to be broadly distributed in the secretory pathway. It is unclear why they would have a role in triggering ROP secretion, given this broad distribution. It suggests that the phenotype of failed ROP secretion is indirect, perhaps due to another protein that fails to traffic correctly to the apical end? MIC15 and TSP1 appear to be good candidates for this function since they lose apical localization on knockout of CRMPB. It would be informative to test if evacuoles are also disrupted by knockout of MIC15 and/or TSP1. The authors do not adopt this conclusion but rather favor a more complicated, and less well supported, conclusion that the extracellular domains of CRMPs are involved in signaling. If that were the case, the truncations that lack the extracellular domain might be informative. In the absence of such data, I do not see a compelling argument for rejecting the more direct conclusion that CRMPA/CRMPB are part of the secretory pathway and are required for trafficking of MIC15 and TSP1, which are in turn required for ROP secretion. 

4) The material presented in Figure S7 is highly speculative and premature. I agree that domain predictions shown in Figure 6A are interetsing, but they only comprise a small portion of the mature CMPR proteins and hence cannot be used to infer their overall structure much less function. I would suggest removing these speculative sections for future studies that might provide greater mechanistic support for the claims.

5) It is unclear why the authors choose to name the Tg orthologs CRMPA and CRMPB when in other apicomplexans they are numbered. This seems likely to lead to confusion.

Reviewer #3: Singer et al. provide compelling evidence for a novel invasion complex in Toxoplasma gondii that may answer the longstanding question of how rhoptry secretion is triggered. Prior to publication, I recommend that the reviewers address the following concerns:

Major Comments

1. The claim that TSP1 and MIC15 function in a complex with CRMPA and CRMPB requires further investigation to be convincing. They appear less important for plaque formation and invasion than CRMPA and CRMPB, and their expression and localization are not impacted by loss of CRMPB (unlike CRMPA). E-vacuole formation (rhoptry secretion) was not tested for TSP1 and MIC15 either. These data suggest they could be in a separate, nearby complex. One suggestion to strengthen this claim could be to do the reciprocal co-IP with detection of binding partners by immunoblotting (i.e. pull down TSP1 and MIC15 separately and immunoblot for CRMPA and CRMPB). Otherwise, in vitro binding experiments could be performed with recombinant proteins.

Minor comments

1. Invasion doesn't need to be capitalized in title.

2. Abstract: "Here we demonstrate that both proteins form a complex that contains additional micronemal proteins". This statement is ambiguous as it could be interpreted that CRMPA and CRMPB are micronemal proteins, which it appears that they are not.

3. Intro: "at least two addition" to "at least two additional".

4. Fig 1A: tree appears cut off on left side, the outgroup with 6 accessions (e.g. DICDI Q86I19) looks disconnected. Indicate color coding in legend, "colored by apicomplexan species".

5. Fig 1B: aa numbers are unnecessarily small and illegible.

6. Consider using US/UK period for decimal points instead of comma for this journal (e.g. "phenotypic score of -3.95 and -3.64 respectively".

7. Explain the general functions of Kringle, GCC2, and GCC3 domains in the intro, results or discussion.

8. Results: "To determine if these proteins are delivered to the plasma membrane," to "To determine if CRMPB is delivered to the plasma membrane," . sYRP2-CRMPB but not sYFP2-CRMPA was tested in Fig 2C.

9. Fig 2: Results beg the question, is CRMPA and CRMPB on the surface of extracellular parasites or only displayed following contact with host cells or microneme secretion?

10. Fig 3A: since plaque area wasn't quantified, discuss apparent growth defect caused by floxing CRMPA (in absence of rapamycin as compared to the parental line). Does floxing CRMPA reduce basal levels or cause alternative splicing?

11. Results (Fig 3C): state whether CRMPA and CRMPB were confirmed to be knocked out in the vacuoles counted for replication.

12. Fig 4A: "massspec" to "mass spec"

13. Results: "While deletion of TGG1_285000" to "While deletion of TGGT1_285000"

14. Results: "Similarly, deletion of tetratricopeptide protein (TGG1_247000)" to "Similarly, deletion of tetratricopeptide proteins (TGGT1_247000)"

---

## [Decision Letter · Decision Letter 2]

14 Nov 2022

Dear Dr. Singer,

Thank you for your patience while we considered your revised manuscript "A central protein complex essential for invasion in Toxoplasma gondii" for publication as a Research Article at PLOS Biology. This revised version of your manuscript has been evaluated by the PLOS Biology editors, the Academic Editor and the original reviewers.

Based on the reviews and our Academic Editor's assessment of your revision, we are likely to accept this manuscript for publication, provided you satisfactorily address the remaining points raised by the reviewers. We encourage you to cite this manuscript (https://www.embopress.org/doi/full/10.15252/embj.2022111158) as one that was published as this was in the final stages of revision.

Please also make sure to address the following data and other policy-related requests.

1. DATA POLICY:

A) Supplementary files (e.g., excel). Please ensure that all data files are uploaded as 'Supporting Information' and are invariably referred to (in the manuscript, figure legends, and the Description field when uploading your files) using the following format verbatim: S1 Data, S2 Data, etc. Multiple panels of a single or even several figures can be included as multiple sheets in one excel file that is saved using exactly the following convention: S1_Data.xlsx (using an underscore).

B) Deposition in a publicly available repository. Please also provide the accession code or a reviewer link so that we may view your data before publication.

Regardless of the method selected, please ensure that you provide the individual numerical values that underlie the summary data displayed in the following figure panels as they are essential for readers to assess your analysis and to reproduce it: Figures 1AB, 2E, 3BCE, 4BCE, 5BDH, 6ABCDE and supplementary figures S1AB, S3A, S5A.

**Please also ensure that figure legends in your manuscript include information on where the underlying data can be found, and ensure your supplemental data file/s has a legend.**

We require the original, uncropped and minimally adjusted images supporting all blot and gel results reported in an article's figures or Supporting Information files. We will require these files before a manuscript can be accepted so please prepare and upload them now. We require this for figure S2E.

Please carefully read our guidelines for how to prepare and upload this data: https://journals.plos.org/plosbiology/s/figures#loc-blot-and-gel-reporting-requirements

3. We recommend a change in the title to include the protein complex: "A central CRMP protein complex is essential for invasion in Toxoplasma gondii".

We expect to receive your revised manuscript within two weeks. We recommend to send the revised manuscript by the due date, as we think it would be important for this manuscript to be published in 2022.

*Published Peer Review History*

*Press*

Sincerely,

Paula

---

Senior Editor,

pjaureguionieva@plos.org,

PLOS Biology

Reviewer remarks:

Reviewer #1: The authors did the experiments suggested by the reviewers and also incorporated the results, explanations, and discussions into the text. I do not have further comments. Congratulations to all authors!

Reviewer #2: I would like to thank the authors for their thorough replies and for addition of new data to support their conclusions. I think many of the points claimed here are well supported including: 1) CRMPA and CRMPB interact with each other, 2) CMRPs are widely distributed in the secretory pathway and partially overlap with MIC proteins, although they are not themselves contained in micronemes, 3) CRMPA primary interacts with MIC15 while CRMPB primarily interacts with TSP1, 3) All 4 proteins form a complex that is necessary for ROP secretion. The one claim that I am not convinced by is the conclusion that CRMP1 and CRMPB are on the surface and that they might require this location for function. The data presented in Figure 2 is by the authors own estimation "suggestive" (lines 194, 200) and even if it does indicate that a small portion of CRMPA and CRMPB reach the surface it does not show that this location is critical to their function. Additionally, the loss of CRMPs results in mislocalization of MIC15 and TSP1 so that they are now retained in the ER. The resulting mislocalization could explain the phenotypes of both the CRMP conditional knockdowns and the MIC15 and TSP1 knockdowns. As the current data cannot distinguish between these models, they need to soften the claims about CRMPs functioning at the surface and add this alternative explanation to the discussion.

Reviewer #3: Singer et al. have adequately addressed all of my concerns in the revised version of this manuscript. Upon review, no additional concerns were noted in the revised manuscript.

---

## [Editor Report · Decision Letter 3]

29 Nov 2022

Dear Dr Singer,

Thank you for the submission of your revised Research Article "A central CRMP protein complex essential for invasion in Toxoplasma gondii" for publication in PLOS Biology. On behalf of my colleagues and the Academic Editor, Kami Kim, I am pleased to say that we can in principle accept your manuscript for publication, provided you address any remaining formatting and reporting issues. These will be detailed in an email you should receive within 2-3 business days from our colleagues in the journal operations team; no action is required from you until then. Please note that we will not be able to formally accept your manuscript and schedule it for publication until you have completed any requested changes.

PRESS

Sincerely, 

Paula

---

Senior Editor

PLOS Biology
